# Hypothalamic Rax$^+$ tanycytes contribute to tissue repair and tumorigenesis upon oncogene activation in mice

Wenhui Mu[1,8], Si Li [1,2,8], Jingkai Xu [1,3,8], Xize Guo[1,2], Haoda Wu[1,2], Zhenhua Chen[1,2], Lianyong Qiao[1], Gisela Helfer[4], Falong Lu [1,2], Chong Liu[5] & Qing-Feng Wu [1,2,6,7✉]

Hypothalamic tanycytes in median eminence (ME) are emerging as a crucial cell population that regulates endocrine output, energy balance and the diffusion of blood-born molecules. Tanycytes have recently been considered as potential somatic stem cells in the adult mammalian brain, but their regenerative and tumorigenic capacities are largely unknown. Here we found that Rax+ tanycytes in ME of mice are largely quiescent but quickly enter the cell cycle upon neural injury for self-renewal and regeneration. Mechanistically, Igf1r signaling in tanycytes is required for tissue repair under injury conditions. Furthermore, Braf oncogenic activation is sufficient to transform Rax+ tanycytes into actively dividing tumor cells that eventually develop into a papillary craniopharyngioma-like tumor. Together, these findings uncover the regenerative and tumorigenic potential of tanycytes. Our study offers insights into the properties of tanycytes, which may help to manipulate tanycyte biology for regulating hypothalamic function and investigate the pathogenesis of clinically relevant tumors.

[1] State Key Laboratory of Molecular Development Biology, Institute of Genetics and Developmental Biology, Chinese Academy of Sciences, Beijing, China. [2] University of Chinese Academy of Sciences, Beijing, China. [3] Department of Dermatology, China-Japan Friendship Hospital, Beijing, China. [4] School of Chemistry and Biosciences, University of Bradford, Bradford, UK. [5] Department of Pathology and Pathophysiology, School of Medicine, Zhejiang University, Hangzhou, Zhejiang, China. [6] CAS Center for Excellence in Brain Science and Intelligence Technology, Shanghai, China. [7] Chinese Institute For Brain Research, Beijing, China. [8] These authors contributed equally: Wenhui Mu, Si Li, Jingkai Xu. ✉email: wu_qingfeng@genetics.ac.cn

Hypothalamic tanycytes share many common features with ependymal cells but display a unique morphology and distinct functional features[1,2]. They are polarized cells, with cell bodies lining the third ventricle, elongated processes extending into the parenchyma, and endfeet contacting the pial surface of the brain[3,4]. In the median eminence (ME), a structural link between the hypothalamus and pituitary gland, tanycytes contribute to the regulation of multiple hypothalamic functions including the diffusion of blood-borne molecules, neuroendocrine output, energy balance, and reproductive ageing[1–3,5–10]. Tanycytes in ME have been implicated in the maintenance of hypothalamus-mediated body homeostasis, but have also been recently identified as a possible key component of stem cell niche due to their prominent expression of neural progenitor markers and their potency of proliferation and differentiation[6–8,11–14]. While somatic stem cells, in general, are characterized by their self-renewal, regenerative, and tumorigenic potential[15–17], it remains largely unknown how ME tanycytes maintain themselves and to what extent the disturbance of such homeostasis contributes to diseases such as cancer.

Single-cell RNAseq technology has recently been applied to dissect the self-renewal and regenerative capacity of ependymal cells[18,19]. In particular, the combination of cell lineage tracing with single-cell profiling shows that ependymal cells do not function as latent neural stem cells (NSCs) to self-renew and repair the damaged neural tissue[19]. As specialized ependymal cells, the role of tanycytes in tissue repair is still unknown. Interestingly, the ultrastructural analysis identifies tanycytes as uniciliated ependymal cells[20], implicating a potentially different function from typical ependymal cells.

Another important characteristic of somatic stem cells is their tumor-initiating potential[15,21,22], but the tumorigenic capacity of tanycytes has never been reported. Craniopharyngioma is a benign but aggressive intracranial tumor that occurs in between the hypothalamus and pituitary gland[23–25]. Clinically, craniopharyngioma is subdivided into adamantinomatous and papillary subtypes, which differ in genetic mutations and onset of age[24,25]. Papillary craniopharyngiomas are restricted to adult-onset with a much higher mortality rate and frequently driven by somatic Braf$^{V600E}$ mutation[24–27]. Although the clinical endoscopic studies imply that papillary craniopharyngiomas may originate from the hypothalamus-pituitary axis including ME[28,29], the cell-of-origin for adult-onset craniopharyngiomas remains unknown. We wondered whether ME tanycytes could develop into craniopharyngioma-like tumors upon oncogene activation in the adult brain.

Retina and anterior neural fold homeobox transcription factor (Rax) is selectively expressed in hypothalamic tanycytes, especially those at the ventral part of the third ventricle[30]. Here, we aim to deconstruct the regenerative and tumorigenic potential of Rax$^+$ tanycytes and speculate that whether tanycytes in ME serve as cells-of-origin for craniopharyngioma. We thereby manipulated tanycyte biology and performed lineage tracing using Rax-CreER$^{T2}$ knock-in mice, and revealed that Rax$^+$ tanycytes responded to neural injury for regeneration and contributed to tumorigenesis upon Braf oncogene activation.

## Results

**Transcriptomic analysis of tanycytes in ME and their niche cells.** Adult ME has been reported to represent a robust proliferative structure, encompassing a 15-fold higher density of dividing cells compared to other hypothalamic regions[14,31]. Recently, a molecular census of arcuate nucleus and ME cells focused on the heterogeneity of neurons and their roles in regulating energy homeostasis[32]. However, the mitotic activity of tanycytes in the niche remains contentious. To profile the transcriptome of tanycytes in ME (containing β2 subtype) and its neighboring hypothalamic niche (Fig. 1a), we microdissected this particular region from adult mice and acutely dissociated the cells for single-cell RNAseq (Supplementary Fig. 1; "Methods"). Our unsupervised analysis of 990 qualified cells clearly discriminated ten distinct clusters (Fig. 1b). Using expression patterns of cell-type-specific marker genes[32], we assigned a single identity to each cluster: tanycytes (Rax$^+$), vascular and leptomeningeal cells (VLMCs, Lum$^+$), erythrocytes (Hba$^+$, Hbb$^+$), neurons (Syp$^+$), oligodendrocyte precursor cells (OPCs, Cspg4$^+$), pars tuberalis cells (PTCs, Slc47a1$^+$, Tshb$^+$), natural killer cells (NK cells, Nkg7$^+$), macrophages/microglia (Aif1$^+$), and oligodendrocytes (Mog$^+$) (Supplementary Fig. 2a–e and Supplementary Data 1). The putative stem cell markers Sox2, Slc1a3, Vimentin (Vim), and Aldoc were enriched but not exclusively expressed in tanycytes (Supplementary Fig. 2d). Moreover, tanycytes shared a set of stemness-associated genes with NSCs and ependymal cells (Supplementary Fig. 2f and Supplementary Data 2)[19].

**Adult Rax$^+$ tanycytes are largely quiescent.** Actively dividing cells in ME have been reported to control systemic energy balance, but whether ME tanycytes are mitotically active remain controversial[12,14,31]. To determine their mitotic activity, we first confirmed Rax as a molecular marker for ME tanycytes[30,32] and revealed Scn7a, Col25a1, and Mia as specific markers (Fig. 1c), confirming the recent census of cell types in the hypothalamus[32,33]. The molecular signatures were validated by single-molecule fluorescent in situ hybridization (smFISH) and Rax-driving lineage tracing, showing the spatial distribution of tanycytes in both ependymal (EZ) and subependymal zone (SEZ) (Fig. 1d–f and Supplementary Fig. 2g). Next, we mapped the expression of cell cycle-related genes (e.g., Mki67 and Mcm2) using single-cell transcriptomic analysis and unexpectedly found that the mitotic genes were predominantly enriched in OPCs but rarely expressed in ME tanycytes (Supplementary Fig. 3a, b).

To reveal the identity of dividing cells, we labeled mitotically active cells by intraperitoneal ethynyldeoxyuridine (EdU) administration in adult mice and costained EdU-positive cells with cell-specific markers such as Rax (tanycytes), Sox2 (NSCs), NG2 (OPCs), Olig2 (OPCs and oligodendrocytes), NeuN (neurons), CC1 (oligodendrocytes), S100β (astrocytes), or Iba1 (microglia) in the hypothalamus. Our results showed that Rax$^+$ tanycytes did not undergo active cell division (Fig. 1d). Further analyses uncovered that up to 60% of labeled dividing cells expressed Sox2, in accordance with the ratio of Olig2$^+$ cells, and approximately 40% of mitotic cells were positive for NG2 (Fig. 1g, h and Supplementary Fig. 3c, d). Given the striking coexpression of NG2 (encoded by Cspg4) or Olig2 with Sox2 revealed by both immunostaining and single-cell transcriptomic analysis (Supplementary Fig. 3e–h), we conclude that Sox2$^+$Olig2$^+$ OPCs rather than Sox2$^+$Olig2$^-$ tanycytes predominate the mitotically active cells in ME.

Furthermore, we traced the cell lineage of tanycytes using the Rax-CreER$^{T2}$::Rosa26-Stop-tdTomato (also known as Ai14) mouse line (Supplementary Fig. 4a). We tested the specificity of instant genetic labeling at 1 day post tamoxifen induction (dpi) by smFISH and found that 99% of tdTomato$^+$ cells in ME were positive for tanycyte marker Scn7a (Fig. 2a), despite the sparse leaky labeling of cells with neuronal morphology in the hypothalamic nuclei (Supplementary Fig. 4b). The cell fate of Rax$^+$ tanycytes was then analyzed at 3, 7, 14, and 30 days following a single dose of tamoxifen injection. We first analyzed the spatial distribution of induced cells and found that the starter cells included ~80% of ventricular cells along EZ and 20% of parenchymal cells in SEZ (Fig. 2b). Notably, the proportion of cells distributed in EZ and SEZ did not change during 30 days of lineage

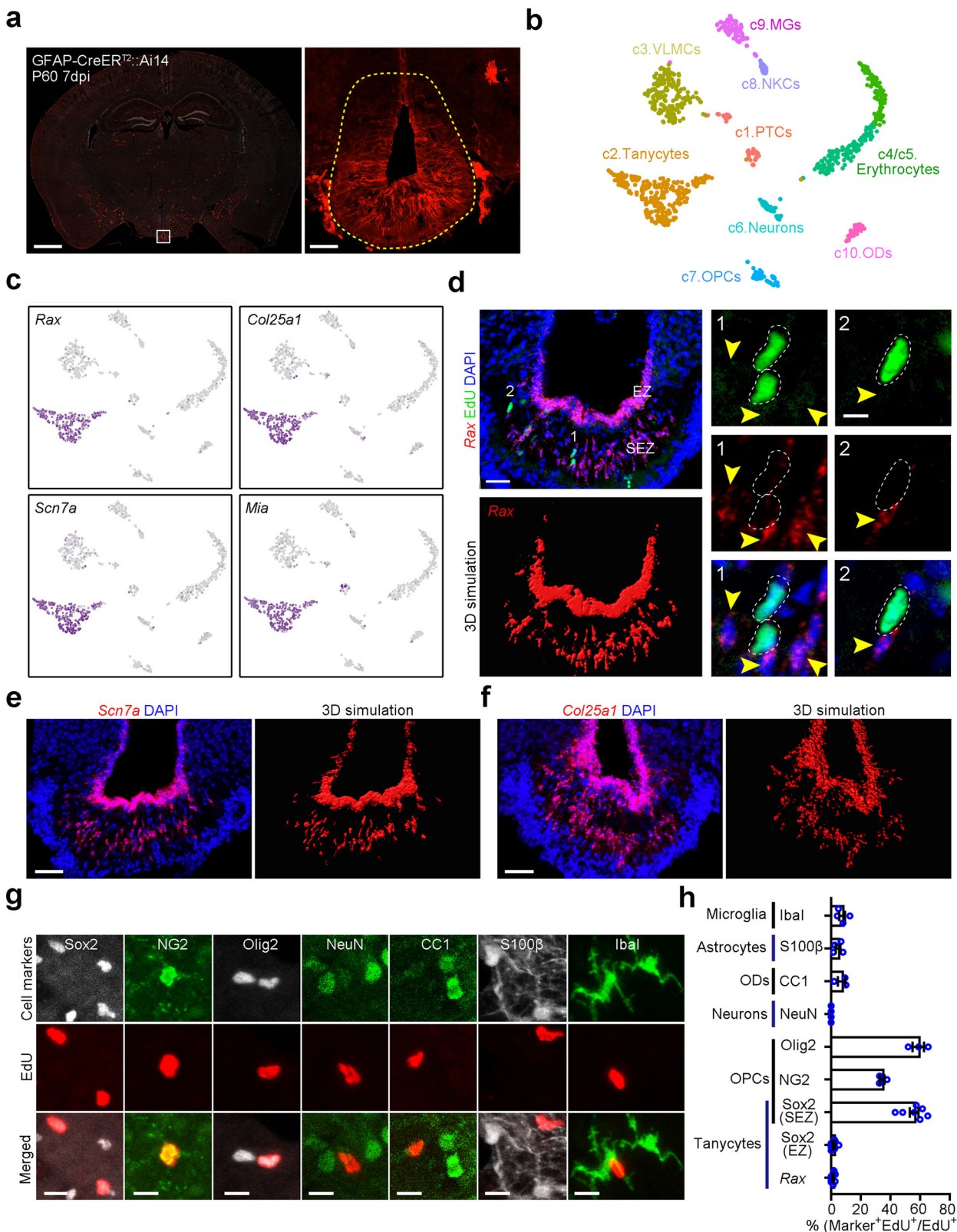

tracing (Fig. 2b, c). Second, we assessed the identity of traced cells with multiple cell-specific markers and found that Rax$^+$ tanycytes did not generate neurons, oligodendrocytes and astrocytes under physiological conditions (Fig. 2d, e). Lastly, we combined lineage tracing with EdU labeling assay to examine whether tanycytes divide and self-renew. Our data showed that cell division of traced tanycytes was not detectable, or very rare (Fig. 2f, g). Collectively, these data

support our hypothesis that Rax$^+$ tanycytes are relatively quiescent under physiological conditions.

**Rax$^+$ tanycytes respond to neural injury.** It has been reported that adult ependymal cells are postmitotic and remarkably stable without any proliferative potential, even after stroke-induced

**Fig. 1 Tanycytes in ME are slow-cycling cells. a** Representative image of fluorescent immunoreactivity in adult coronal brain sections from GFAP-CreER[T2]::Ai14 mice. White box indicates the magnified view of the median eminence (ME) on the right. The mice were induced with tamoxifen at postnatal day 60 (P60) and sacrificed at 7 days post induction (dpi) to guide microdissection of ME. Scale bars: 1000 μm (left) and 50 μm (right). **b** Spectral t-distributed stochastic neighbor embedding (tSNE) representation of the single-cell transcriptomic data with clusters colored and annotated according to known cell-type markers. PTCs pars tuberalis cells, VLMCs vascular and leptomeningeal cells, OPCs oligodendrocyte precursor cells, NKCs natural killer cells, MGs microglia, ODs oligodendrocytes. **c** tSNE scatter plots covering neural cell clusters show the specific expression of *Rax*, *Col25a1*, *Scn7a* and *Mia* in tanycytes. **d** Representative images of smFISH probing for *Rax* mRNA display no colocalization of EdU with *Rax*. Numbers refer to subregions shown in the magnified images (right). Yellow arrowheads indicate Rax+ tanycytes and dashed white circles signify EdU+ mitotic cells. EZ ependymal zone, SEZ subependymal zone. Scale bar: 40 μm (left) and 10 μm (right). **e, f** Shown are *Scn7a* and *Col25a1* gene expression in ME detected by smFISH and three-dimensional (3D) simulation of *Scn7a*- and *Col25a1*-positive signals. Scale bars: 50 μm. **g** Representative images stained for EdU with multiple cell markers including Sox2, NG2, Olig2, NeuN, CC1, S100β, and Iba1. Three daily EdU administrations were applied to label dividing cells. Scale bars: 10 μm. **h** Quantification of the percentage of different cell types among mitotic cells. Values represent mean ± SEM (n = 7, 7, 7, 3, 3, 4, 3, 4, and 4 mice from bottom to top bars).

neural injury[19,34]. As specialized ependymal cells, whether tanycytes are involved in the repair of local injury is not clear. To investigate the regenerative capacity of tanycytes, we induced a mechanical injury by penetrating an acupuncture needle into ME (Fig. 3a). In contrast to the quiescence of tanycytes in sham control mice, mechanical injury caused a robust activation of Scn7a+ tanycytes in both EZ and SEZ (Fig. 3b). Cell-type analysis of dividing cells revealed that the number of cycling tanycytes (Sox2+Olig2− or Scn7a+), OPCs (Sox2+Olig2+ or PDGFRα+) and astrocytes (S100β+) were strikingly increased after injury (Fig. 3c, d and Supplementary Fig. 5a). The results of fate mapping using Rax-CreER[T2]::Ai14 mice further showed that neural injury induced a predominant self-renewal of tanycytes and drove a small number of tanycytes to differentiate into OPCs or astrocytes (Fig. 3e and Supplementary Fig. 5b–d), implicating the multipotential differentiation of activated tanycytes into glial cells. Notably, given that our heat-based antigen retrieval approach deteriorated the tdTomato signal to a certain extent, whether ME tanycytes could robustly differentiate into OPCs or astrocytes upon injury requires further confirmation.

We then induced targeted neural injury via genetic cell ablation of tanycytes in Rax-CreER[T2]::Rosa26-Stop-diphtheria toxin receptor (DTR)−2A-GFP (iDTR) mice (Supplementary Fig. 5e–g), whereby the tamoxifen and diphtheria toxin dosage was optimized to avoid a complete ablation of tanycytes. Subsequently, we evaluated the mitotic activity of cells in the damaged tissue and found that targeted neural injury also induced a significant increase in the number of dividing cells (Fig. 3f and Supplementary Fig. 5h). The further cell-type assessment revealed that tanycytes self-renewed following induced cell ablation, indicating that residual tanycytes are capable of regenerating to compensate for their cell loss (Fig. 3f–h). Quantitative analysis confirmed that neural injury increased the number of cycling oligodendrocyte lineage cells and astrocytes (Fig. 3i). We also tracked the fate of Rax+ tanycytes after genetically-induced injury and found that a very small number of tanycytes could transform into astrocytes or OPCs (Supplementary Fig. 5i). Nevertheless, neurogenesis in ME was very limited even after neural injury (Fig. 3i). These results demonstrate that Rax+ tanycytes expand themselves with environmental insult for tissue damage repair.

**Igf1r signaling is required for tanycyte preservation and injury-induced tissue repair.** Given that tanycytes possess regenerative capacity, it is important to investigate the regulatory mechanism of their preservation and tanycyte-mediated tissue repair. To determine the molecular basis for maintaining tanycyte population, we assessed the expression of receptor tyrosine kinases (RTKs) in our single-cell transcriptomic dataset and found that insulin-like growth factor 1 receptor (*Igf1r*) and neurotrophic receptor tyrosine kinase 2 (*Ntrk2*) were enriched in

tanycytes among the 51 RTKs (Supplementary Fig. 6a). Further analysis showed that the expression of Igf1r in tanycytes was relatively specific, compared to epithelial growth factor receptors (EGFRs), fibroblast growth factor receptors (FGFRs), platelet-derived growth factor receptors (PDGFRs), and neurotrophic receptors (Fig. 4a, b).

To investigate whether signaling through Igf1r is critical for preserving the number of tanycytes, we used Rax-CreER[T2]::Igf1r[f/f] mice to delete *Igf1r* in adult tanycytes by applying tamoxifen at P60 and examined the cell population of tanycytes at 2 months post genetic deletion. Our data showed that the population of tanycytes (Sox2+Olig2− or Scn7a+) in SEZ was remarkably declined by loss of Igf1r, but there was no change in OPC number (Fig. 4c, d and Supplementary Fig. 6b, c). Further analysis showed that the deficiency in Igf1r signaling increased the number of apoptotic cells in ME (Supplementary Fig. 6d, e) and pharmacological inhibition of Igf1r signaling in cultured neural progenitors subtly compromised the expression level of stemness-related protein Sox2 (Supplementary Fig. 6f–h). Collectively, these findings suggest that Igf1r signaling is critical for maintaining the tanycyte population.

We further induced a mechanical injury in Igf1r-deficient mice to observe the role of Igf1r signaling in neural tissue repair (Fig. 4e). The results revealed that genetic ablation of *Igf1r* in tanycytes significantly reduced the number of cycling tanycytes and OPCs in damaged ME (Fig. 4f, g), indicating that inhibition of Igf1r signaling impairs tissue regeneration by tanycytes.

**Adult Rax+ tanycytes can contribute to tumor formation.** Beyond the regenerative ability, somatic stem cells have frequently been found to generate tumors upon targeted with oncogenic mutations. We, therefore, determined to investigate the tumorigenic potential of Rax+ tanycytes. Considering the specific anatomic location of craniopharyngioma in between the hypothalamus and pituitary[25,35], we wonder whether Rax+ tanycytes in ME contribute to its formation. A vast majority of adult-onset craniopharyngiomas have been reported to harbor the oncogenic Braf[V600E] point mutation, which is rarely found in meningioma and ependymoma[25,26,36–38]. We introduced an inducible mutant Braf into tanycytes using Rax-CreER[T2]::Braf[V600E] mice and observed the fate of tanycytes at 1 week post tamoxifen induction (Fig. 5a). Strikingly, the constitutive activation of Braf quickly incited the mitotic activity of tanycytes in both EZ and SEZ (Fig. 5b, c). Further cell-type mapping analysis revealed that Braf activation enhanced the expansion of tanycytes per se but did not promote their differentiation into neurons and astrocytes in the short-term (Fig. 5d–f).

Furthermore, we examined the pathology of ME at 2 months post induction (mpi) of Braf[V600E] expression and found the formation of tumor bearing *Braf* mutation at the floor of the third

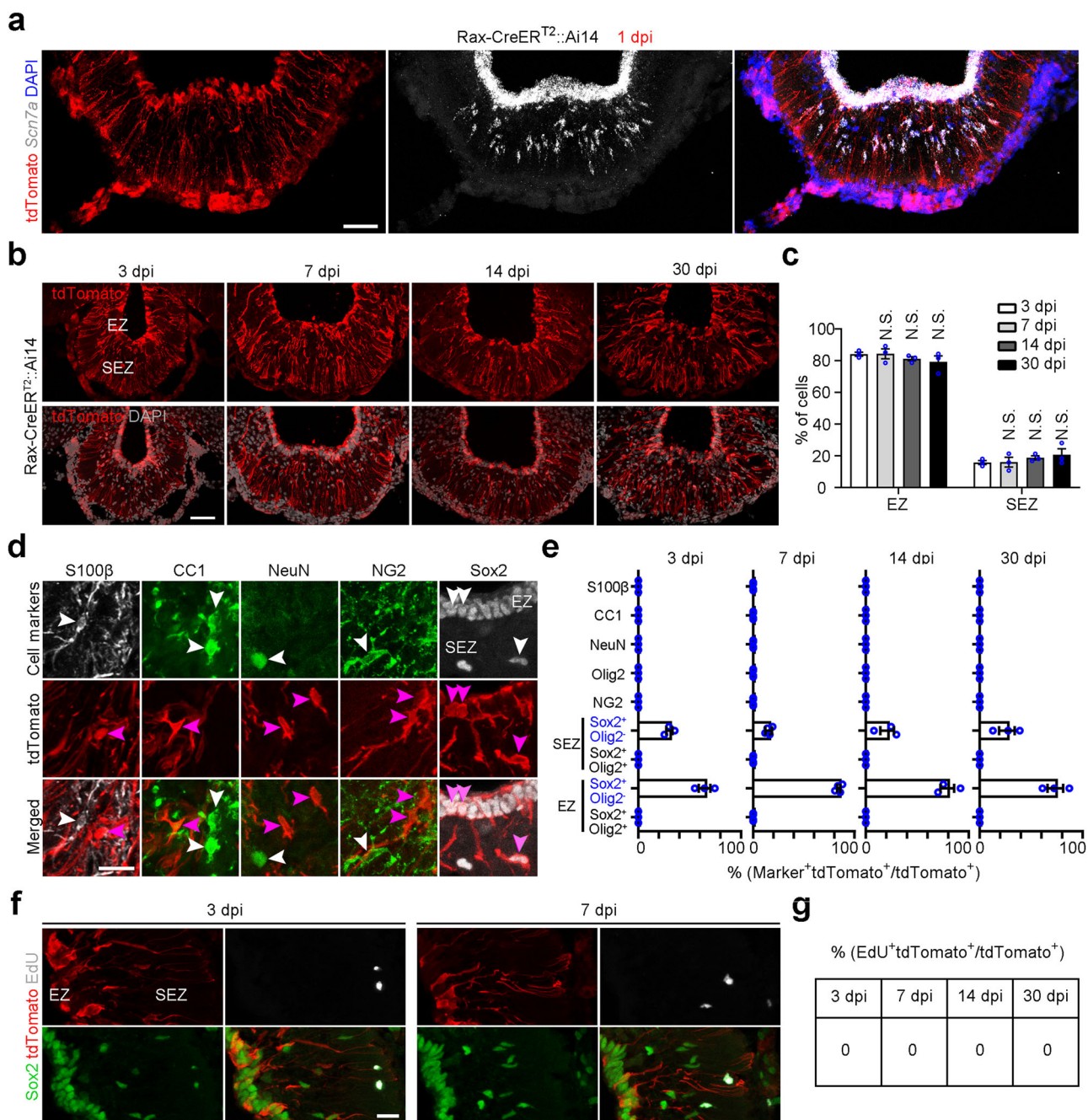

**Fig. 2 Arrest of Rax+ tanycyte lineage during homeostasis. a** Representative images showing that tdTomato+ cells from Rax-CreER$^{T2}$::Ai14 mice are positive for tanycyte marker *Scn7a* at 1 dpi with tamoxifen. The *Scn7a* mRNA was detected by smFISH. Scale bar: 50 μm. **b** Lineage tracing of Rax+ tanycytes in ME at the population level using Rax-CreER$^{T2}$::Ai14 mouse line. Shown are the confocal images of tdTomato+ cells in ME at 3, 7, 14, and 30 dpi with tamoxifen. Scale bar: 50 μm. **c** Quantification of the percentage of tdTomato+ cells distributed in EZ and SEZ. Values are presented as mean ± SEM, and significance was analyzed by one-way ANOVA with Sidak's multiple comparison test (*n* = 3 biologically independent animals). N.S. not significant versus 3 dpi. **d** Representative confocal images obtained from fate mapping of Rax+ tanycytes show the costaining of tdTomato with different cell markers Sox2, NG2, NeuN, CC1, and S100β. White arrowheads indicate marker-positive cells and purple arrowheads signify tdTomato+ cells. Scale bar: 20 μm. **e** Quantification of the percentage of tdTomato+ cells expressing different cell markers. Values represent mean ± SEM (*n* = 3, 4, 3, and 3 mice for 3, 7, 14, and 30 dpi groups). **f** Shown are representative confocal images stained for Sox2, tdTomato, and EdU in ME of Rax-CreER$^{T2}$::Ai14 mice. Scale bar: 20 μm. **g** Percentage of EdU+ mitotic cells among tanycytes. Source data are provided as a Source Data file. The precise *P* values are summarized in Supplementary Data 3.

ventricle (Fig. 6a and Supplementary Fig. 7a, b). The pathological features of induced tumors in ME resembled papillary craniopharyngioma[27]. The width and height of ME were significantly enlarged in the tanycyte-specific mutant mice (Fig. 6b). To validate whether the neoplasm originates from

tanycytes, we generated Rax-CreER$^{T2}$::Braf$^{V600E}$::Ai14 mice and induced oncogene activation during adulthood. As expected, these animals developed identifiable, well-circumscribed tumors which were clearly positive for tdTomato and Vimentin (Fig. 6c and Supplementary Fig. 7c), suggesting the origin of papillary

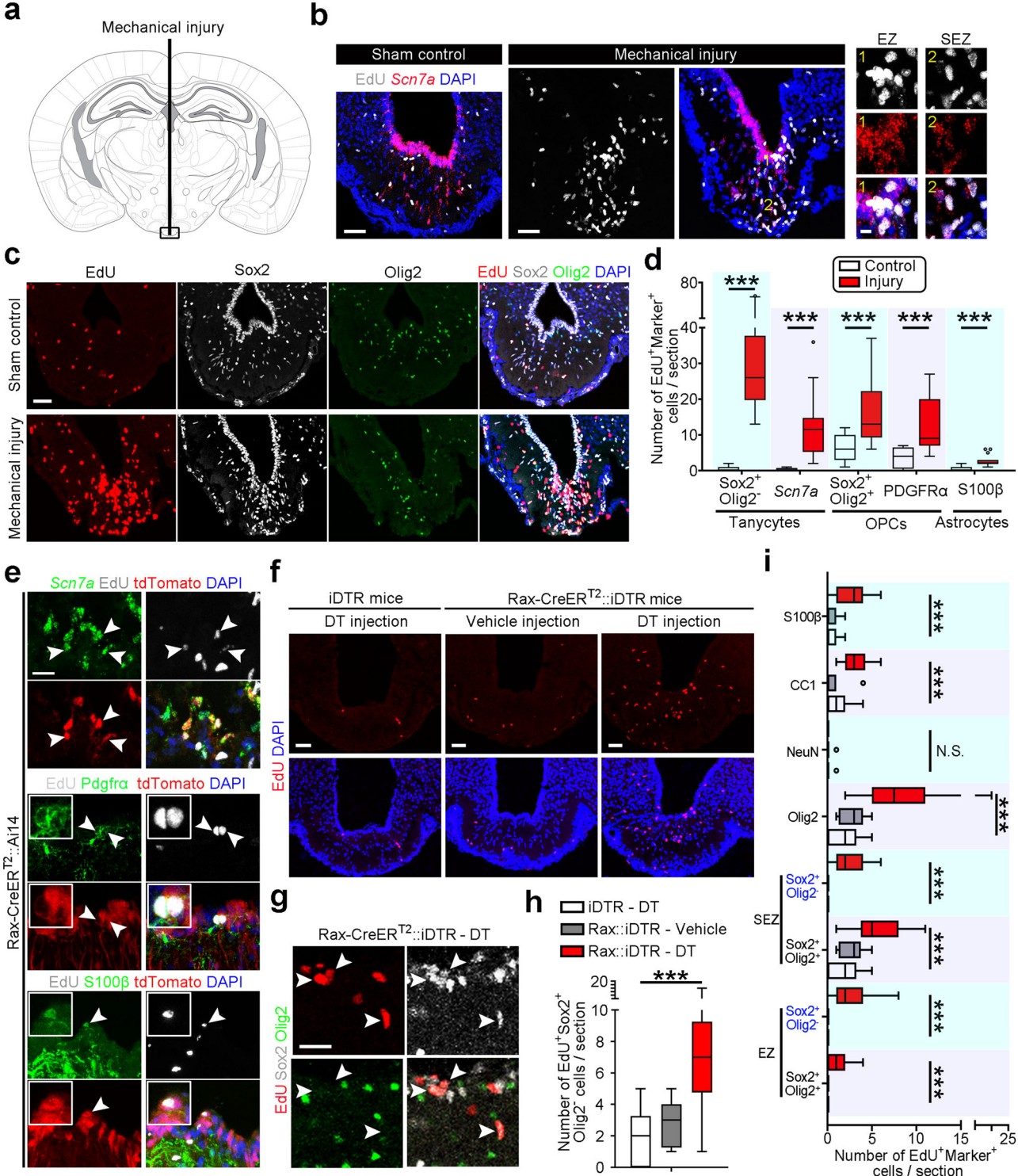

craniopharyngioma-like neoplasm from Rax$^+$ tanycytes. We next assessed the expression pattern of cell-specific markers in the center and marginal zone of tumors generated in the Rax-CreER$^{T2}$::Braf$^{V600E}$ model. Importantly, we observed that the tumor cells in the center zone displayed a prominent enrichment of stem cell markers Sox2 and Sox9, but did not express differentiated cell markers Olig2, CC1, S100β, and PDGFRα as observed in oligodendrocytoma and/or astrocytoma (Fig. 6d, e and Supplementary Fig. 7d–f). The cell division detected by EdU incorporation and Ki67 staining occurred in both the central and peripheral zone of the tumor (Fig. 6d, e and Supplementary

Fig. 7e). Our quantitative analysis showed that the center zone of tumor tissue mainly consisted of tanycyte-like cells but lacked multiple neural lineage cells, while glia-like cells contributed to the cell expansion in the tumor marginal zone (Fig. 6f). A substantial enhancement of OPC division and differentiation was also observed in the neighbor region of tumor tissues (Fig. 6g), implying the potential activation of OPCs in proximity to tumor tissue.

To characterize the malignancy and invasiveness of papillary craniopharyngioma-like tumor, we further performed xenograft transplantation, histological analysis of multiple organs

**Fig. 3 Rax$^+$ tanycytes participate in injury-induced regeneration. a** A schematic diagram of mechanical injury in ME using acupuncture needle. The coronal atlas figure was adapted from "The Mouse Brain in Stereotaxic Coordinates"[57]. **b** Representative images of smFISH targeting *Scn7a* mRNA show colocalization between EdU and Scn7a after mechanical injury. Numbers refer to subregions shown in the magnified images. Scale bars: 50 μm (left) and 10 μm (right). **c** Sample confocal images stained for Sox2, Olig2, EdU, and DAPI in ME of control and injured mice. Three daily EdU injections were applied to label mitotic cells before tissue collection. Scale bar: 50 μm. **d** Quantification of the mitotic cell number of tanycytes (Sox2$^+$Olig2$^-$ or Scn7a$^+$), OPCs (Sox2$^+$Olig2$^+$ or PDGFRα$^+$), and astrocytes (S100β$^+$) in control and injured mice. Boxes represent the interquartile range (IQR), whiskers extend to ±1.5 IQR and bold black lines indicate the median values ($n = 19, 10, 16, 12, 19, 10, 13, 15, 21$, and 9 sections from at least three mice from left to right boxes). Statistical significance was analyzed by unpaired two-tailed Student's $t$ test. **$P < 0.01$; ***$P < 0.001$. **e** Representative confocal images showing the triple labeling of tdTomato, EdU and diverse cell markers in the injured ME, suggesting that mechanical injury induces the self-renewal of tanycytes (tdTomato$^+$EdU$^+$Scn7a$^+$) and their differentiation into OPCs (tdTomato$^+$EdU$^+$PDGFRα$^+$) and astrocytes (tdTomato$^+$EdU$^+$S100β$^+$). White boxes show the magnified view of cells pointed by the arrowheads. The adult tanycytes in Rax-CreER$^{T2}$::Ai14 mice were labeled by a single injection of tamoxifen, followed by mechanical injury at 7 dpi. The mice received three daily EdU injection at 1 day after injury and were then sacrificed for smFISH analysis or immunostaining. Scale bars: 20 μm. **f** Mitotic activity detected by EdU labeling in iDTR mice treated with diphtheria toxin (DT) and Rax-CreER$^{T2}$::iDTR (abbreviated as Rax::iDTR) mice receiving vehicle or DT injection. Scale bar: 50 μm. **g** Immunostaining images showing the mitotic activation of Sox2$^+$Olig2$^-$ tanycytes after genetic ablation-mediated injury. Arrowheads indicate EdU$^+$Sox2$^+$Olig2$^-$ tanycytes. Scale bar: 10 μm. **h** Quantification of the number of Sox2$^+$Olig2$^-$ tanycytes undergoing cell division after targeted neural injury. Boxes represent IQR, whiskers extend to ±1.5 IQR, and significance was analyzed by one-way ANOVA with Sidak's multiple comparison test ($n = 22, 12$, and 18 sections from at least three mice from left to right boxes). ***$P < 0.001$. **i** Quantitative analyses showing the expansion of tanycytes, oligodendrocyte lineage cells, and astrocytes after partial tanycyte ablation. Boxes represent IQR, whiskers extend to ±1.5 IQR, dots represent outliers, and significance was analyzed by one-way ANOVA with Sidak's multiple comparison test ($n = 22, 12, 18, 22, 12, 18, 22, 12, 18, 22, 12, 18, 22, 12, 18, 22, 12, 18, 13, 10, 12, 20, 10, 14, 22, 12$, and 19 sections from at least three mice from bottom to top boxes). ***$P < 0.001$; N.S. not significant. Source data are provided as a Source Data file. The precise $P$ values are summarized in Supplementary Data 3.

susceptible to tumor metastasis and in vitro culture of tumor cells. Our data showed that tumor cells derived from Rax-CreER$^{T2}$::Braf$^{V600E}$ mice did not cause neoplasm in naked mice or proliferate massively in vitro, and neither did the neoplasm invade into other organs at 6 mpi (Supplementary Fig. 8).

We then dissected the mouse tumor tissues, constructed cDNA libraries, and performed bulk RNAseq. Principal component analysis revealed that the mRNA expression profile of tumor tissues was distinguishable from control ME tissues (Fig. 7a). After applying a standard filtering approach that compared tumor with normal tissues, we identified 291 differentially expressed genes (Fig. 7b–d). The tumor tissues displayed an enriched expression of transcription factors (e.g., *Dlx2*, *Gbx2*, *Foxb1*, and *Nkx2.2*) critical for brain development, tanycyte markers (e.g., *Ptprz1*, *Fabp7*, *Crym*, and *Gja1*), and differentiation markers (e.g., *Dcx*, *Olig1*, and *Cspg5*) (Fig. 7d). The gene ontology analysis of tumor-enriched genes demonstrated that the transcriptional programs involved in cell division, forebrain, diencephalon, and midbrain development were upregulated in the papillary craniopharyngioma model (Fig. 7e and Supplementary Fig. 9a). Surprisingly, genes engaged in immune response were down-regulated in tumor tissues (Fig. 7f and Supplementary Fig. 9b). Together, our results suggest that tanycytes may serve as a cell-of-origin for papillary craniopharyngioma in the adult brain and use the transcriptional programs activated during brain development to evolve into tumor cells.

## Discussion

Manipulation of tanycyte biology could provide a valuable tool for regulating hypothalamic function and modeling disease. Understanding the property, plasticity, and potential of tanycytes is fundamentally important to maintain their stemness, enhance their function and restrain their tumorigenesis. Here we show that Rax$^+$ tanycytes in ME robustly transit from a quiescent to an active state for tissue regeneration when subjected to neural injury. Loss of Igf1r signaling in tanycytes leads to a substantial decrease of tanycyte population and impairs tissue repair after injury. Importantly, Rax$^+$ tanycytes are susceptible to Braf oncogene activation and display tumorigenic potential (Fig. 8).

A combination of bioinformatics analysis, lineage tracing, and pulse-chase assay demonstrate that Rax$^+$ tanycytes are largely quiescent, supporting previous studies showing that β tancytes

residing in ME cannot proliferate but NG2 glia are actively dividing cells there[12,31]. Nevertheless, adult tanycytes are not permanently dormant like ependymal cells in lateral ventricles[19]. While αSMA$^+$ ependymal cells fail to activate under growth factor infusion and striatal injury condition[19], targeted neural injury can induce robust cell division of Rax$^+$ tanycytes, suggesting them as latent hypothalamic stem cells with the regenerative capacity to replenish cell loss and repair injured tissues. Despite that our data have shown that a minority of tanycytes, if not none, differentiate into OPCs or astrocytes by dividing, we still could not exclude the possibility of transdifferentiation of tanycytes into glial cells. How residual or neighbor tanycytes sense the damage in their microenvironment and regenerate to maintain tissue homeostasis is still unclear. A recent study shows that interferon γ signaling within the injury niche promotes the exit of NSCs from dormancy into a primed state[39], providing a potential mechanism underlying the activation of quiescent NSCs following neural injury.

By datamining the single-cell transcriptomic database, we identify *Igf1r* as a specifically expressed RTK in tanycytes and reveal that the Igf1 signaling pathway is required for tissue repair after injury. Genetic deletion of *Igf1r* in Rax$^+$ tanycytes compromises their own maintenance and regenerative capacity. These findings provide a potential druggable target to preserve tanycyte population during aging and promote tissue regeneration after traumatic brain injury.

Somatic stem cells not only maintain their own stemness and tissue homeostasis after injury but also possess the potential of oncogenic transformation[15,21,22]. Previous reports suggest that Braf$^{V600E}$ expression in neural progenitors is not sufficient for tumorigenesis and needs to cooperate with *Ink4a* or *Cdkn2a* locus deficiency in oligodendrocyte or astrocyte precursor cells to induce glioma formation[40–43]. Unexpectedly, a remarkable result reported here is the discovery that an introduction of mutant Braf$^{V600E}$ into tanycytes is sufficient to drive tumor formation at the floor of the third ventricle, demonstrating that tanycytes have tumorigenic potential and are susceptible to *Braf* mutation. Craniopharyngioma is a heterogeneous brain tumor of uncertain origin and its tumor biology is poorly understood[25]. Recent studies show that pituitary stem/progenitor cells carrying *Cnntb1* but not *Braf* mutation contribute to the development of pituitary tumors and/or adamantinomatous craniopharyngioma via

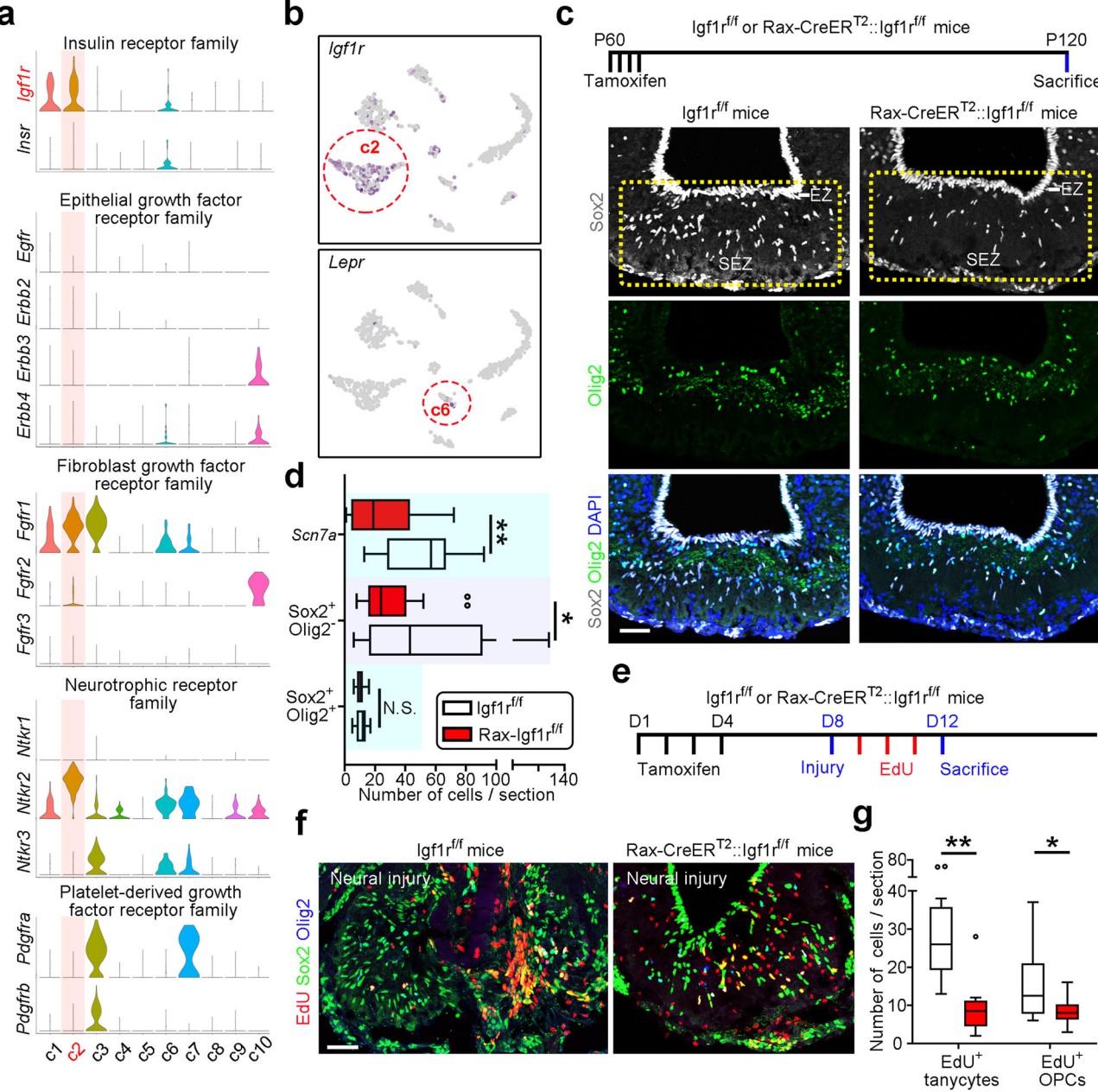

**Fig. 4 Igf1r signaling is required for preserving tanycytes and tissue repair. a** Violin plots showing the expression of the insulin receptor family, epithelial growth factor receptor family, fibroblast growth factor receptor family, and platelet-derived growth factor receptor family in ME tanycytes, and the microenvironmental cells. The clusters are distinguished by different colors. **b** tSNE scatter plots showing that Igf1r is enriched in the tanycyte population but Lepr is undetectably low in ME. Cluster c2 represents tanycytes, and c6 represents neurons in ME. **c** Top, experimental scheme. Bottom, representative confocal images immunostained for Sox2, Olig2, and DAPI in the ME of control and Igf1r conditional knockout mice at 2 months post tamoxifen injection. Scale bar: 50 µm. **d** Quantification of the number of $Scn7a^+$ or $Sox2^+Olig2^-$ tanycytes and $Sox2^+Olig2^+$ OPCs in control and Igf1r-deficient mice. Boxes represent IQR, whiskers extend to ±1.5 IQR and dots represent outliers ($n = 21, 20, 21, 20, 12$, and 20 sections from at least three mice from bottom to top boxes). P values were analyzed using unpaired two-tailed Student's t test. **e** Experimental scheme of mechanical injury in the ME of control and Igf1r conditional knockout mice followed by EdU labeling. **f** Sample confocal images stained for Sox2, Olig2, and EdU in injured ME of control and Igf1r-deficient mice. Scale bar: 50 µm. **g** Quantification of the mitotic cell number of tanycytes and OPCs in control and Igf1r-deficient mice. Boxes represent IQR and whiskers extend to ±1.5 IQR ($n = 12$ and 10 sections from three independent animals for Igf1r^{f/f} and Rax-Igf1r^{f/f} groups, respectively). P values were analyzed using unpaired two-tailed Student's t test. *$P < 0.05$, **$P < 0.01$; ***$P < 0.001$; N.S. not significant. Source data are provided as a Source Data file. The precise P values are summarized in Supplementary Data 3.

paracrine mechanism[44–46]. Interestingly, our results reveal that tanycyte-derived tumors in ME mimic papillary craniopharyngioma (frequently carrying Braf^{V600E} mutation) with respect to their anatomic location, genetic mutation, and pathological features, suggesting that hypothalamic Rax^+ tanycytes could serve as

a cell-of-origin for papillary craniopharyngioma. In contrast to the salient OPC features in glioma[47,48], the tumor cells in our disease model do not abundantly express differentiated cell markers such as Olig2, S100β, and CC1. Notably, craniopharyngioma was previously assumed to derive from squamous

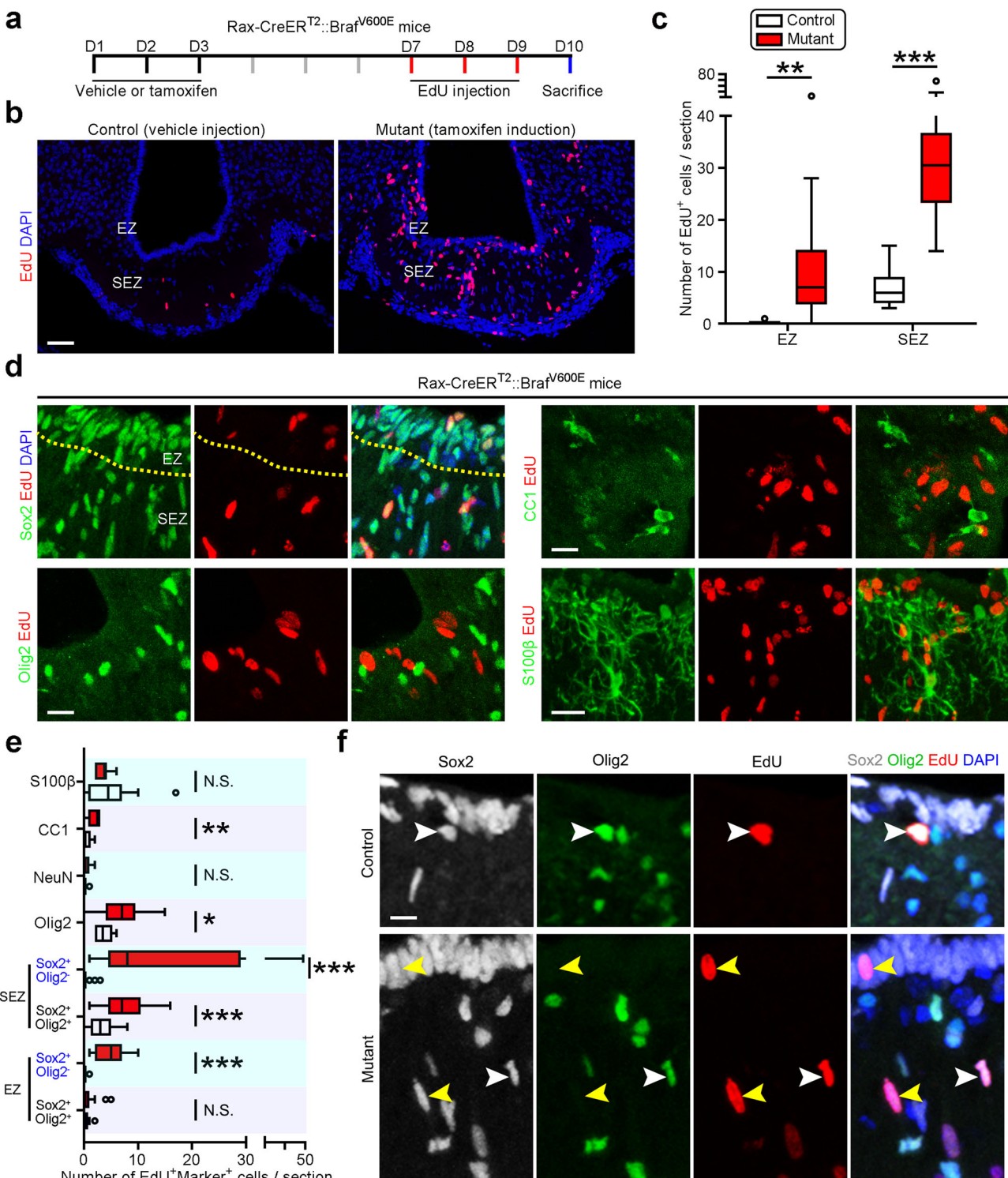

**Fig. 5 Introduction of Braf$^{V600E}$ stimulates mitotic activation of Rax$^+$ tanycytes. a** Experimental scheme describing the induction of somatic genetic mutation in tanycytes using Rax-CreER$^{T2}$::Braf$^{V600E}$ mice and detection of cell division in ME. **b** Sample confocal images showing the mitotic cells in the ME of control and Braf-mutant mice. Scale bar: 50 μm. **c** Quantification of EdU$^+$ cell number in EZ and SEZ of control and mutant mice. Boxes represent IQR, whiskers extend to ±1.5 IQR, and significance was analyzed by unpaired two-tailed Student's *t* test (*n* = 12 and 14 sections from three mice for control and mutant groups). ***$P < 0.001$. **d** Representative confocal images stained for EdU with Sox2, Olig2, CC1, and S100β in Rax-CreER$^{T2}$::Braf$^{V600E}$ mice receiving tamoxifen injection. Scale bar: 20 μm. **e** Quantification of the number of mitotic cells expressing different cell markers (*n* = 20 sections from four mice). Boxes represent IQR, whiskers extend to ±1.5 IQR and significance was analyzed by unpaired two-tailed Student's *t* test (*n* = 20, 21, 20, 21, 20, 21, 20, 21, 10, 17, 18, 10, 10, 10, 18, and 10 sections from at least three mice from bottom to top boxes). *$P < 0.05$; ***$P < 0.001$; N.S. not significant. **f** Sample images showing mutant Braf$^{V600E}$ activates mitotic division of Sox2$^+$Olig2$^-$ tanycytes in ME. White arrowheads signify Sox2$^+$Olig2$^+$ OPCs while yellow arrowheads indicate tanycytes. Scale bar: 10 μm. Source data are provided as a Source Data file. The precise *P* values are summarized in Supplementary Data 3.

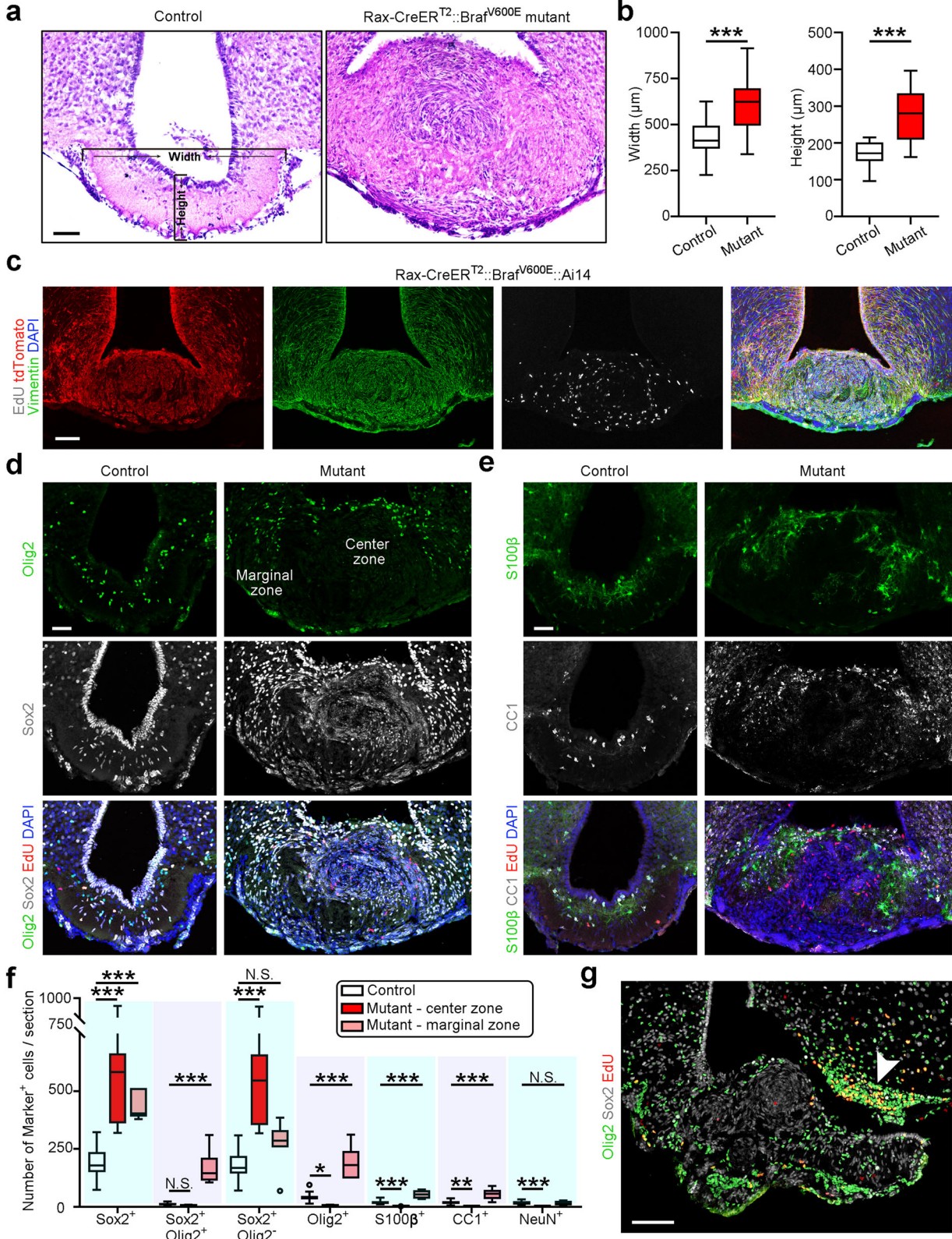

epithelial cells by clinical researchers[25], but we did not find a robust transcriptomic similarity between tanycytes and esophageal squamous epithelia, or between tanycytes-derived tumor and esophageal squamous cell carcinoma (Supplementary Fig. 10). The data from our transcriptomic profiling further demonstrate that tanycyte-derived tumors display enrichment of genes involved in the brain but not pituitary development. Due to the

lack of transcriptomic data of human papillary craniopharyngioma in previous studies, further investigation is required to uncover the similarity and difference between mouse and human tumor tissues.

Given that previous studies have subdivided tanycytes into α1, α2, β1, and β2 subtypes along the third ventricle[5,32], here we have to emphasize that our study focuses on Rax⁺ tanycytes,

**Fig. 6 Rax$^+$ tanycytes may serve as a cell-of-origin of papillary craniopharyngioma. a** Hematoxylin and eosin (HE) staining of ME in control and Rax-CreER$^{T2}$::Braf$^{V600E}$ mutant mice at 2 months post tamoxifen injection. Scale bar: 50 μm. **b** Quantification of the width and height of ME in control and mutant mice. Boxes represent IQR, whiskers extend to ±1.5 IQR, and significance was analyzed by unpaired two-tailed Student's *t* test (*n* = 24 and 7 sections from four mice for control and mutant groups). ***$P$ < 0.001. **c** Representative confocal images showing the costaining of tdTomato, Vimentin, and EdU in Rax-CreER$^{T2}$::Braf$^{V600E}$::Ai14 mice at 2 months post tamoxifen injection. Scale bar: 100 μm. **d**, **e** Representative confocal images of tumor tissues stained for Olig2, Sox2, S100β, CC1, EdU, and DAPI. Three daily EdU injections were applied to label dividing cells before tissue collection. Scale bars: 50 μm. **f** Quantitative analyses show the number of cells expressing different markers in control tissues, center zone, and marginal zone of tumors. Sox2$^+$Olig2$^-$ cell number is increased in the center zone, while the number of Sox2$^+$Olig2$^+$ OPCs is augmented in the marginal zone of tumors. Boxes represent IQR, whiskers extend to ±1.5 IQR, and significance was analyzed by one-way ANOVA (*n* = 24, 7, 7, 24, 7, 7, 24, 7, 7, 24, 7, 7, 21, 7, 7, 21, 7, 7, 21, 7, and 7 sections from four mice from left to right boxes). Given that tumor cells expanded locally, we only had a limited number of brain sections from neoplastic tissues for quantification. ***$P$ < 0.001; N.S. not significant. **g** Sample confocal image showing the expansion of OPCs in the arcuate nucleus in proximity to ME. Scale bar: 100 μm. Source data are provided as a Source Data file. The precise $P$ values are summarized in Supplementary Data 3.

predominantly composed of β1 and β2 subtypes. However, the regenerative and tumorigenic potentials of α tanycytes remain elusive to us.

Taken together, our findings identify Rax$^+$ tanycytes as slow-cycling cells wherein quiescence may preserve their inherent longevity[15]. We further reveal the previously underappreciated role of adult tanycytes in tissue repair and tumorigenesis and establish a craniopharyngioma mouse model, which may unlock an avenue for regenerative medicine and cancer therapy in the central nervous system.

## Methods

**Animals**. All experimental mice were under the husbandry care of the Institute of Genetics and Developmental Biology, Chinese Academy of Sciences. The mouse strains including Rax-CreER$^{T2}$ (strain name: Raxtm1.1(cre/ERT2)Sbls/J; stock number: 025521), Igf1r$^{f/f}$ (strain name: B6;129-Igf1rtm2Arge/J; stock number: 012251), Braf$^{V600E}$ (strain name: Braftm1Mmcm/J; stock number: 017837), Ai14 (strain name: B6.Cg-Gt(ROSA)26Sortm9(CAG-tdTomato)Hze/J; stock number: 007914), iDTA (strain name: B6;129-Gt(ROSA)26Sor$^{tm1(DTA)Mrc}$/J; stock number: 010527), and GFAP-CreER$^{T2}$ (strain name: B6.Cg-Tg(GFAP-cre/ERT2)505Fmv/J; stock number: 012849) were obtained from the Jackson Laboratory. Inducible DTR mouse line (strain name: B6-Gt(ROSA)26Sortm2(CAG-LSL-DTR-EGFP)) was generated by Biocytogen. Two-month-old wild-type C57BL/6N mice and 5-week-old BALB/c-nu mice were ordered from SPF Biotechnology Co. Ltd and housed in the same animal facility on a 12-h reverse light/dark cycle and provided with food and water ad libitum. The animal facility was maintained at a temperature of 21 °C with 50–60% humidity. All mice in the study were backcrossed to the C57BL/6N background for at least six generations. To perform lineage tracing, targeted neural injury and cell-type-specific genetic manipulation, we crossed Rax-CreER$^{T2}$ driver mice with Ai14, iDTR, Igf1r$^{f/f}$, or Braf$^{V600E}$ mice during adulthood to generate corresponding mouse lines. All procedures, husbandry, and experiments were performed according to the policies and ethical regulations established by the Institutional Animal Care and Use Committee at the Institute of Genetics and Developmental Biology, Chinese Academy of Sciences. The experimental protocols used in this study received ethical approval from the Chinese Academy of Sciences.

**Single-cell library preparation and sequencing**. Median eminence (ME) tissues were dissected from ten female adult C57BL/6N mice and pooled together. Single-cell RNA sequencing was performed on 10×Genomics platform. Library was prepared with the Chromium Single Cell 3' Library & Gel Bead Kit v2 (PN-120237), Chromium Single Cell 3' Chip kit v2 (PN-120236), and Chromium i7 Multiplex Kit (PN-120262) according to the manufacturer's instructions. The library was sequenced on the Illumina NovaSeq6000 platform as follows: 26 bp (Read1) and 98 bp (Read2). Due to the tininess of ME tissue and the lack of experience in dissociating such tiny tissues, we only acquired 990 qualified cells for single-cell analyses. Despite the low number of collected cells, we still provided high-quality data.

**Alignment and quantification of scRNA-seq data**. Cell Ranger v2.1 was used to demultiplex raw base call (BCL) files generated by Illumina sequencers into FASTQ files and perform the alignment, barcode counting and unique molecular identifiers counting. Ensembl BioMart version 84 was used to generate the reference genome.

**Single-cell RNAseq analysis**. The downstream analysis was run with the Seurat (version 3.0) R package. The first step is to normalize the data by a global-scaling normalization method that normalizes, multiplies a scale factor (10,000), and log-transforms the feature expression. Next, features were selected to exhibit high cell-

to-cell variation in the aME dataset by Seurat "FindVariableGenes" command with the following parameters: mean.function equal to ExpMean, dispersion.function equal to LogVMR, x.low.cutoff equal to 0.02, x.high.cutoff equal to 8, and y.cutoff equal to 0.5. The number of aME feature is 4844, accordingly. After a linear scaling processing, aME was used for the calculation of the PCs using Seurat's "RunPCA" command. The final decision on the number of 3D t-distributed stochastic neighbor embedding (tSNE) transformation was made by JackStraw statistics with a sharp drop-off in significance. The statistics were estimated with 100 replicate samplings. The proportion of the data that were randomly permuted for each replicate was set to 1%. Differentially expressed features in each cluster could be calculated with "FindAllMarkers" with the threshold set to 0.25. Finally, we visualized marker expression by DoHeatmap, VlnPlot and FeaturePlot. The molecular signatures of neural stem cells (NSCs) and ependymal cells were identified by reanalyzing the published dataset from cells lining SVZ using Seurat R package[19]. The shared molecular markers among NSCs, ependymal cells and tanycytes were analyzed with OmicsBox software 1.4.

To systematically analyze the mitotic activity of cells in ME, we selected cell cycle-related genes for gene expression analysis as previously described[49], including 43 S-phase genes (*Mcm5, Pcna, Tyms, Fen1, Mcm2, Mcm4, Rrm1, Ung, Gins2, Mcm6, Cdca7, Dtl, Prim1, Uhrf1, Mlf1ip, Hells, Rfc2, Rpa2, Nasp, Rad51ap1, Gmnn, Wdr76, Slbp, Ccne2, Ubr7, Pold3, Msh2, Atad2, Rad51, Rrm2, Cdc45, Cdc6, Exo1, Tipin, Dscc1, Blm, Casp8ap2, Usp1, Clspn, Pola1, Chaf1b, Brip1, E2f8*) and 54 G2/M-phase genes (*Hmgb2, Cdk1, Nusap1, Ube2c, Birc5, Tpx2, Top2a, Ndc80, Cks2, Nuf2, Cks1b, Mki67, Tmpo, Cenpf, Tacc3, Fam64a, Smc4, Ccnb2, Ckap2l, Ckap2, Aurkb, Bub1, Kif11, Anp32e, Tubb4b, Gtse1, Kif20b, Hjurp, Cdca3, Hn1, Cdc20, Ttk, Cdc25c, Kif2c, Rangap1, Ncapd2, Dlgap5, Cdca2, Cdca8, Ect2, Kif23, Hmmr, Aurka, Psrc1, Anln, Lbr, Ckap5, Cenpe, Ctcf, Nek2, G2e3, Gas2l3, Cbx5, Cenpa*).

**Genomic DNA extraction and genotyping**. Genomic DNA extraction from mouse tails was conducted using the standard ethanol precipitation protocol. The tissue digestion solution contains 1 mg/mL of proteinase K, 50 mM Tris-HCl (pH = 8.0), 100 mM EDTA, 100 mM NaCl, and 1% SDS. Genotyping was performed with PCR-based assays using purified genomic DNA and primer pairs to detect Rax-CreER$^{T2}$, GFAP-CreER$^{T2}$, iDTR, iDTA, Igf1r$^{f/f}$, Braf$^{V600E}$, and Ai14 alleles, respectively. The primers used for genotyping are listed in Supplementary Table 1.

**Validation of Braf$^{V600E}$ activation**. We microdissected the neoplastic tissues from Rax-CreER$^{T2}$::Braf$^{V600E}$ mouse model at 2 months post induction (mpi), isolated the mRNA, and performed reverse transcription to obtain cDNA. First, we referred to the source paper reporting the generation of Braf$^{V600E}$ mouse line[50], amplified the target DNA fragment and verified the activation of Braf using XbaI restriction enzyme (NEB, R0145) which cut the recognition site introduced by Dankort et al.[3]. Second, we amplified the target fragment from cDNA and sent the samples for Sanger sequencing to test the mutant oncogene expression in neoplastic tissues.

**EdU, tamoxifen, and diphtheria toxin (DT) administration**. For tracking cell division in hypothalamus and its neighbor niche, adult mice were intraperitoneally injected with EdU (Ark Pharm, AK163060-1g) for three consecutive days (50 mg/kg body weight), followed by tissue collection to analyze the mitotic activity in the hypothalamus at 24 h after the last injection.

To induce genetic recombination in Rax-CreER$^{T2}$::Ai14, Rax-CreER$^{T2}$::iDTR, Rax-CreER$^{T2}$::iDTA::Ai14, Rax-CreER$^{T2}$::Igf1r$^{f/f}$, Rax-CreER$^{T2}$::Braf$^{V600E}$, and Rax-CreER$^{T2}$::Braf$^{V600E}$::Ai14, the adult mice received a single or multiple intraperitoneal injections of tamoxifen (Sigma-Aldrich, T5648) at the dose of 132 mg/kg body weight. A stock of tamoxifen (66 mg/mL; Sigma; T5648) was prepared in a 5:1 ratio of corn oil to ethanol at 37 °C with occasional vortexing as previously described[51].

To genetically ablate a subpopulation of tanycytes and induce targeted neural injury, we applied daily DT (Sigma-Aldrich, D0564) at the dose of 10 μg/kg for

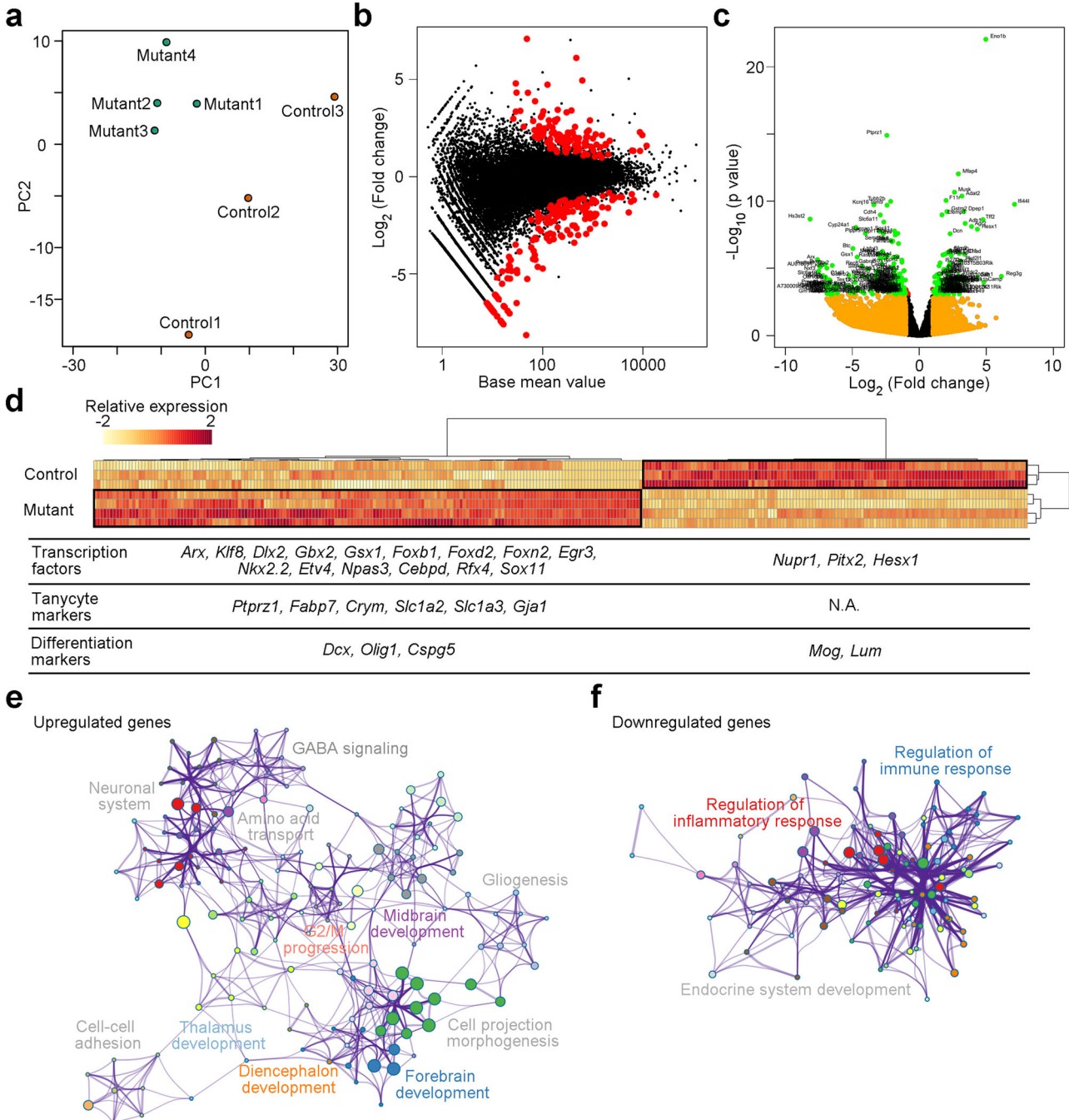

**Fig. 7 Bulk RNAseq analysis of tumor tissues. a–c** Principal component (PC) analysis (**a**), MA plot (**b**), and volcano plot (**c**) showing the differential gene expression in ME tissues of control and Rax-CreER[T2]::Braf[V600E] mutant mice at 2 months post tamoxifen injection. **d** A heatmap showing the differential gene expression of control and papillary craniopharyngioma-like tumor tissues. A set of upregulated and downregulated transcription factors, tanycyte markers, and differentiation markers in the mouse papillary craniopharyngioma are highlighted in the table. **e**, **f** Gene ontology network analysis of upregulated (**e**) and downregulated (**f**) genes in the tumor tissues. The major clusters with different gene ontology are labeled with different colors.

3 days in adult Rax-CreER[T2]::iDTR mice that had received a single tamoxifen (66 mg/ml) injection 3 days before. The procedure has been described in our previous study[52].

**Tissue section preparation, EdU staining, and immunohistochemistry**. The experimental mice were anesthetized by intraperitoneal injection of 4% chloral hydrate and then transcardially perfused with saline followed by 4% paraformaldehyde (PFA) in phosphate-buffered saline (PBS). Mouse brains were dissected, post-fixed for 4–6 h in 4% PFA at 4 °C, and subsequently cryo-protected in 20% sucrose in PBS for 12 h followed by 30% sucrose for 24 h. Tissue blocks were prepared by embedding in Tissue-Tek O.C.T. Compound (Sakura 4583). The brain sections (20 μm in thickness) were prepared using a cryostat microtome (Leica, CM3050S), dried for 30 min at room

temperature in the dark and stored in −20 °C freezer. For immunostaining, the tissue sections were washed with 1×TBS (pH = 7.4, containing 3 mM KCl, 25 mM Trizma base, and 137 mM NaCl) and pre-blocked with 1×TBS + + (TBS containing 5% donkey serum and 0.3% Triton X-100) for 1 h at room temperature, followed by incubation with primary antibodies diluted in TBS + + overnight at 4 °C. The primary antibodies used in this study included Sox2 (Goat; R&D Systems; #KOY0317071; 1:500), NG2 (Rabbit; Millipore; #3018740; 1:200), Olig2 (Goat; R&D Systems; #UPA0617051; 1:200), NeuN (Mouse; Abcam; #ab104224; 1:1000), HuC/D (Mouse; Molecular Probes; #A21271; 1:1000), CC1 (Mouse; Calbiochem; #OP80; 1:200), S100 (Rabbit; Abcam; #ab868; 1:400), Iba1 (Rabbit; Wako; #019-19741; 1:1000), GFP (Goat; Rockland; #33302; 1:1000), RFP (Rabbit; Rockland; #35055; 1:1000), GFAP (Rabbit; DAKO; #Z0334; 1:1000), Igf1r (Rabbit; CST; #30275; 1:200), Sox9 (Rabbit; Abcam; #ab185966; 1:250);PDGFRα (Goat; R&D Systems; #AF1062; 1:200), Vimentin (Rabbit;

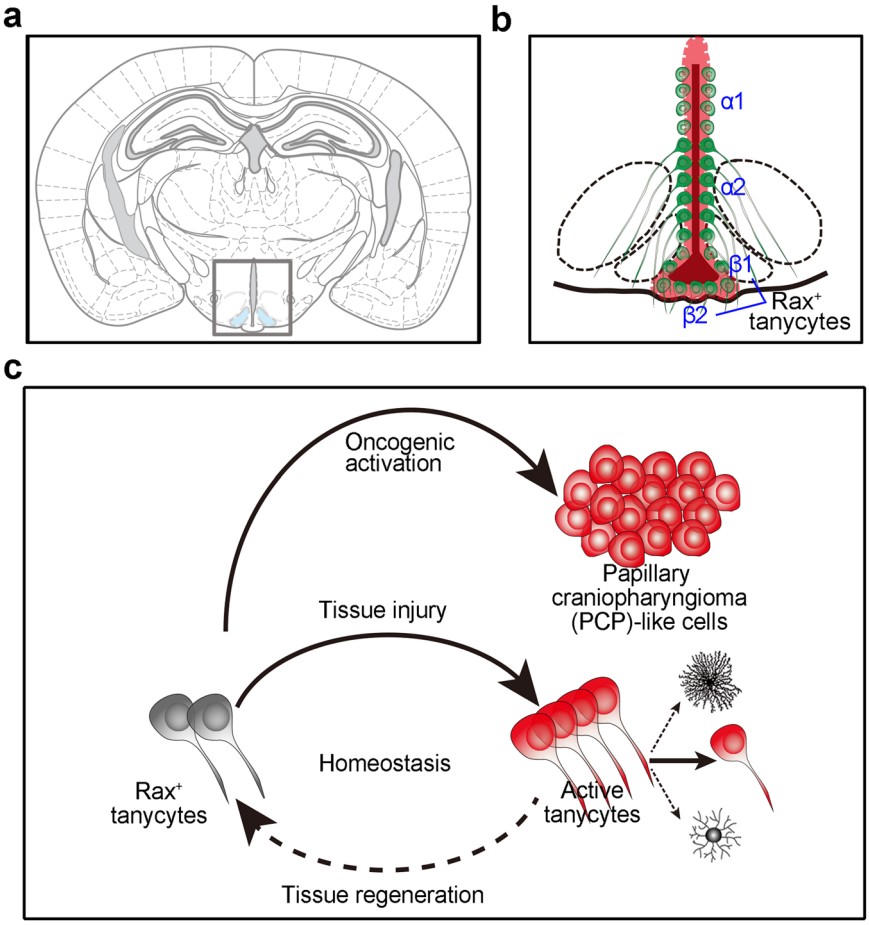

**Fig. 8 A graphical model showing the regenerative and tumorigenic potential of Rax⁺ tanycytes in ME. a** A schematic diagram of coronal mouse brain section, adapted from "The Mouse Brain in Stereotaxic Coordinates"[57]. **b** The spatial distribution of α1, α1, β1, and β2 tanycytes along the 3rd ventricle. **c** Regenerative and tumorigenic potentials of Rax⁺ tanycytes.

Abcam; #ab92547; 1:500), Mia (Goat; R&D; #AF2050, 1:50), Ki67 (Rabbit; Abcam; #ab16667;1:500) and mCherry (Rat; ThermoFish; #SI259077; 1:500). After primary antibody incubation, the brain sections were washed for three times with 1×TBS and incubated with following secondary antibodies for 2 h at room temperature: anti-rabbit/mouse Cy2, anti-goat/rabbit Cy3 and anti-rabbit/goat Cy5 (Donkey; Jackson ImmunoResearch; 1:500).

For EdU labeling, the tissue sections were permeabilized with 0.5% Triton-X-100 for 30 min and detected with detection solution containing 5 μM Sulfo-Cy3/Cy5 azide (Lumiprobe, #C1330/#C3330), 0.1 M Tris-HCl (pH = 7.5), 4 mM copper sulfate, and 100 mM sodium ascorbate for 30 min. After staining, sections were coverslipped with mounting medium, air-dried overnight and maintained at 4 °C in the dark for further imaging. The brain sections were imaged using Zeiss LSM 710 confocal microscopy (Carl Zeiss, Oberkochen, Germany) and Leica SP8 confocal (Leica, Germany) equipped with four lasers (405, 488, 568, and 647 nm). Images were processed and analyzed using Image J software (NIH, Bethesda, MD, USA). For 3D reconstruction, optical stacks from the entire ME were serially aligned along the rostrocaudal axis using Reconstruct 1.1.0 (J.C. Fiala, NIH), followed by import into Imaris 9.0.1 (Bitplane) for further analysis.

**Single-molecule fluorescent in situ hybridization (smFISH).** To prepare tissue sections for smFISH, mouse brains were dissected, immersed in 4% PFA for 4–6 h, and then dehydrated with 20–30% DEPC-treated sucrose for 24 h. Subsequently, the tissues were rapidly frozen on dry ice, embedded in O.C.T. compound, cryosectioned at a thickness of 20 μm, and mounted onto SuperFrost Plus microscope slides. The probes targeting against *Rax* (19046 A), *Scn7a* (18199B), and *Col25a1* (19032A) were designed and validated by Advanced Cell Diagnostics. RNAscope v2 Assay (Advanced Cell Diagnostics, #320511) was used for all smFISH experiments according to the manufacturer's protocol[53]. Briefly, the brain sections were dried at 55 °C for 2 h, rinsed with 1×PBS, treated with 3% hydrogen peroxide in methanol, and subjected to antigen retrieval. Subsequently, the tissue sections were dehydrated with 100% ethanol and incubated with mRNA probes for 2 h at 40 °C. The specific signals were then amplified with multiplexed amplification buffer and detected with TSA Plus fluorophore (Perkin Elmer, #NEL753001KT) for 30 min. To combine smFISH with the pulse-chase assay, we further treated the tissue

sections with 0.5% Triton-X-100 and EdU detection solution. After staining, the slides were mounted and imaged for further analysis.

To detect the mRNA expression level of *Mia*, we used the hybridization chain reaction (HCR) approach[54]. We designed the probe sequences (Supplementary Table 2) using the coding and 3′UTR regions of *Mia* and synthesized the probes in Sangon Biotech, China. The brain sections were permeabilized in 70% ethanol for 16 h at 4 °C, followed by 0.5% Triton X-100 in 1×PBS at 37 °C for 1 h, and treated with 10 μg/mL Protease K was to improve mRNA accessibility. After two washes with 1×PBS at room temperature, sections were pre-hybridized in probe hybridization buffer (30% formamide, 5×SSC, 9 mM citric acid, 0.1% tween 20, 50 μg/mL heparin, 1×Denhardt's solution, and 10% dextran sulfate) for 1 h at 37 °C and then incubated in probe hybridization buffer containing HCR probes (10 μM for each) at 37 °C for 3 h. After mRNA hybridization, the washing and amplification steps were performed as previously described[54].

**TUNEL assay.** For TUNEL assay, cryopreserved sections were fixed in 4% PFA for 20 min at room temperature and incubated with sodium citrate solution containing 0.3% Triton X-100 for 30 min. Subsequently, TUNEL reaction mixture (Roche, 11684795910) was added and incubated with brain sections for 1 h at 37 °C in the dark. After staining, sections were coverslipped with a mounting medium and observed under a microscope.

**Western blotting analysis.** The neural progenitor cells were cultured at 37 °C with 5% $CO_2$ in NeuroCult Basal Medium (STEMCELL, #05700) supplemented with NeuroCult Proliferation Supplement (STEMCELL, #05701), EGF (PeproTech, AF-315-09) and bFGF (PeproTech, AF-450-33). The cells were treated with 1 μM OSI-906 (MedChemExpress, HY-10191) to block Igf1r signaling for 48 h and harvested with ice-cold lysis buffer (20 mM Tris-HCl, pH = 8.0, 100 mM NaCl, 0.5% NP-40, and 1 mM EDTA, 10 mM sodium fluoride, and 5 mM sodium pyrophosphate) containing a cocktail of protease and phosphatase inhibitors (Roche, 04693132001 and 04906837001). The cell lysates were spun down at 20,000×*g* for 20 min and the supernatants were denatured at 95 °C for 10 min. The equivalent denatured samples were subjected to SDS-PAGE, transferred to PVDF membranes

and probed with the antibodies against Sox2 (goat; R&D; AF2018; 1:2000), IGF1R (rabbit; CST; 3027s; 1:1000), phosphorylated IGF1R (rabbit; CST; 3024s, 1:1000) or α-Tubulin (mouse; CST; 3873; 1:5000). Blots were visualized with Super Signal West Pico chemiluminescence (Thermo, 3457) and imaged with a Bio-Rad ChemiDoc™ XRS system. The intensity of stained bands was quantified using Image J software.

**In vitro culture of tumor cells and xenograft transplantation**. The tumor tissues were microdissected from Rax-CreER$^{T2}$::Braf$^{V600E}$ mice at 2 mpi under aseptic conditions and incubated in digestion buffer containing 8 U/mL Papain (Worthington, LK003178) and 100 U/mL DNaseI (Worthington, LK003172) at 37 °C for 30 min. After the termination of digestion with 1×HBSS with 20% fetal bovine serum (FBS), we dissociated the cells with fine-polished glass pipette, spun down the cells at 300×g for 5 min, resuspended the pellets, and filtered the single-cell suspension with a 40-μm cell strainer (BD Falcon, 352340). We used melanoma cells dissected from the skin of Tyr-CreER$^{T2}$::Braf$^{V600E}$::Pten$^{f/f}$ mice (strain name: B6;Cg-Tg(Tyr-cre/ERT2)13Bos Braf$^{tm1Mmcm}$Pten$^{tm1Hwu}$/BosJ, stock number: 013590) as the positive control. For in vitro culture, we transferred the filtered cell suspension to 96-well plates and cultured the cells in DMEM/F12 (Gibco, Life Technologies) with 10% fetal bovine serum and 100 U/mL penicillin/streptomycin, followed by passaging when the tumor cells reached 100% confluency. For xenograft transplantation, a total of 10$^7$ cells suspended in 1×PBS were mixed with the equivalent volume of matrigel (BD, 356234) and then subcutaneously injected to 7-week-old female nude mice. The xenograft tumor growth was monitored over time and imaged for visualization.

**Mechanical injury**. Adult C57BL/6N mice, Rax-CreER$^{T2}$::Ai14 mice, and Rax-CreER$^{T2}$::Igf1r$^{f/f}$ mice were anesthetized with 4% chloral hydrate and subjected to mechanical injury with an acupuncture needle penetrating the midline of brains until reaching ME. The stereotactic penetration of acupuncture needle into ME was applied with the following coordinate: 2 mm posterior to bregma and 7 mm below dura. At 24 h after mechanical injury, the mice were intraperitoneally injected with EdU (Ark Pharm, AK163060-1g) for 3 consecutive days (50 mg/kg body weight), followed by tissue collection to analyze the mitotic activity of cells in ME.

**Histology analysis**. The mouse brains, liver, spleen, lung, and kidney embedded in O.C.T. compound were sectioned at a thickness of 10–20 μm with cryostat microtome (Leica, CM3050S). The tissue sections were washed with 1×PBS buffer, sequentially stained with hematoxylin and eosin (H&E, Beyotime, #C0105), and then dehydrated with gradient ethanol. Subsequently, we clarified the stained tissue sections with xylene and mounted the slides with DPX mounting medium (Sigma, #06522). The images were taken with a Leica SP8 microscopy and Nikon ECL IPSE Ci-L microscope.

**Cell quantification**. To quantify the number of cells expressing cell-type-specific markers, we serially sectioned the ME and analyzed the positive cells in one out of five brain sections spanning the whole ME (from −1.58 mm to −2.3 mm posterior to bregma). Quantification of cells labeled by EdU and expressing different cell-type markers (Sox2, Olig2, NG2, CC1, S100β, and NeuN) in ME was conducted using Carl Zeiss AIM 2.3 software, Leica LAS X software or Image J 1.52p (NIH) software. Generally speaking, at least three animals and five brain sections from each animal were used for quantification. To investigate the spatial distribution of cells, we respectively defined the ependymal layer and the parenchymal region in ME as EZ and SEZ. For neoplastic tissues, we subdivided tumors into center and marginal zone based on the histological morphology and then quantified the number of cells expressing various cell markers. Given that tumor cells expanded locally, we only had a limited number of brain sections (n = 2–3) for quantification.

**Bulk RNAseq of tumor tissues**. To compare the transcriptomic profiles between normal and tumor tissues, we microdissected the ME region from three control and four Rax-CreERT2::Braf$^{V600E}$ mice at 2 months post tamoxifen induction and isolated the total RNAs with Trizol reagent (Ambion). Reverse transcription and template switching were performed with oligo(dT) guided primers and SuperScript II Reverse Transcriptase (Invitrogen, #18064-022) according to the Smart-seq2 protocol[55]. Subsequently, we amplified the cDNA samples with KAPA HiFi Hot-Start polymerase (KAPA Biosystems, KR0369) and purified them with SPRIselect beads (Beckman Coulter, B23318). The cDNA libraries were then prepared using TruePrep DNA Library Prep Kit V2 (Vazyme, TD503-02) and sequenced on an Illumina NovaSeq6000 platform. To compare the transcriptomic profiles of tanycyte- and esophageal squamous cell-derived tumors, we used the recently published bulk RNAseq data collected from normal esophageal epithelial cells and esophageal epithelial cell carcinoma[56], and performed principal component analysis (PCA) and Spearman's correlation analysis.

**Statistics and reproducibility**. All data collection and cell quantification were blinded in this study. Rstudio, GraphPad Prism (v8), and Microsoft Excel 2016 were used to test covariates and normality, calculate statistical significance, and prepare quantitative graphs. Group data are presented as bar plots showing mean ± standard error of the mean (SEM) or box plots wherein boxes represent the interquartile range (IQR), whiskers extend to ±1.5 IQR, dots represent outliers and bold black lines indicate median values. Statistical analysis was calculated using unpaired two-tailed Student's t test or one-way ANOVA with Sidak's multiple comparison test. If not otherwise noted, statistical significance was indicated as follows: *P < 0.05; **P < 0.01; ***P < 0.001; N.S. not significant. The precise P values and sample size were provided as an additional Supplementary Data 3. All results shown in the study are representative of at least two independent experiments with similar results.

**Reporting summary**. Further information on experimental design is available in the Nature Research Reporting Summary linked to this paper.

## Data availability

The single-cell and bulk RNAseq data in this study have been deposited in the Gene Expression Omnibus (GEO) with accession number GSE132943. All other relevant data that support the findings of this study are available within the article and its Supplemental Information files or from the corresponding authors upon reasonable request. A reporting summary for this article is available as a Supplementary Information file. Source data are provided with this paper.

## Code availability

The computational code used in this work is available upon request.

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

## Acknowledgements

We thank Dr. Yinqing Li and Samuel Z.H. Wong for their comments and helpful discussion. The Rosa26-Stop-diphtheria toxin receptor (DTR)−2A-GFP and Rosa26-iDTA mouse lines were kindly provided by Dr. Weixiang Guo and Dr. Yi Zeng, respectively. The work was supported by National Key Research & Development Program of China (2019YFA0801900 and 2018YFA0801104), National Natural Science Foundation of China (32070972, 31771131, 31800860, 81891002, and 31921002), Strategic Priority Research Program of Chinese Academy of Sciences (XDB32020000), Hundred-Talent Program (Chinese Academy of Sciences) and Beijing Municipal Science & Technology Commission (Z181100001518001).

## Author contributions

Q.F.W., W.M., and S.L. designed all experiments. W.M., S.L., and J.X. performed the experiments and statistical analyses. Q.F.W., H.W., Z.C., and X.G. performed bioinformatics analysis of single-cell and bulk transcriptomic datasets. F.L. and L.Q. contributed to the generation of the cDNA library for sequencing from tanycyte-derived tumor tissues. G.H. and C.L. provided technical advice. Q.F.W and S.L. wrote the paper.

## Competing interests

The authors declare no competing interests.
