## [Peer Review File · Nature Communications]

Reviewers' comments:

Reviewer #1 (Remarks to the Author):

The manuscript by Mu and colleagues addresses the proliferative potency of tanycytes in adult mice and their potential role in tumorigenesis. Specifically, they looked into the ability of the median eminence (ME)-lining tanycytes to develop into craniopharyngioma-like tumors upon oncogene activation in the adult brain. By performing single-cell RNAseq analysis of the ME area the authors identified several transcripts that are enriched in tanycytes and confirmed their RNAseq data by single-molecule FISH labeling. Transcriptomic analysis of the cell-cycle mitotic genes revealed that they were predominantly enriched in OPCs but not in the tanycytes. They further confirmed these data by treating mice with mitotic marker EdU and tracing EdU+ cells shortly after injections. The oncogene induced proliferation of tanycytes seems to be the remarkable finding of the study, as a single mutation is sufficient to drive proliferation suggesting that tanycytes perhaps represent an exceptionally vulnerable population for tumor induction.

The major weakness of the study:

- The authors claim that injury or cell ablation induces tanycytic self-renewal and differentiation, and although differentiation was demonstrated by colocalization of reporter (tdTomato) and cell-type markers (S100b or Olig2), self-renewal was never confirmed. To prove the cell-renewal capacity of tanycytes the authors must show triple labeling of their Tomato reporter with a cell-division marker (i.e. EdU) and a tanycyte marker (i.e. Rax, Scn7a, vimentin, etc).

- While the oncogene activation results seem exciting, pointing to tanycytes as tumor source, the authors actually do not provide clear direct evidence in support (although they have tools that could easily demonstrate it). Specifically, it is puzzling for the reviewer that the authors did not cross in a fluorescence (e.g. tdTomato) reporter to the Rax-CreER: BrafV600E model. This way, a tanycytic origin of the tumor cells could be easily proven. Making the case for it just based on the absence of Olig2 from newly formed (EdU labelled) Sox2 cells is too indirect.

Also at many other occasions the authors distinguish tanycytes from OPCs by double-labeling for Olig2 and Sox2, which is often not very convincing. It is unclear why they did not make more use of the specific tanycytic markers (Rax, vimentin, etc).

Other concerns:

1 It would be important to include high-magnification double labeling images for the genes that are claimed as novel tanycytic markers (Scn7a, Col25a1, Mia) with tdTomato.

2 In Suppl. Fig. 5B: S100b/tdTomato double-labeling is somewhat confusing as two arrowheads point to different types of S100b-labeling: the left arrowhead points at a cytoplasmic signal, whereas the right one points at a nuclear signal.

3 Suppl. Fig. 5B indicates DAPI, however the images do not contain a DAPI signal.

4 Suppl. Fig. 5C model is inaccurate as the data do not support it. First, tanycytes have not been demonstrated to self-renew (see above). Secondly, Olig2 labels both OPCs and oligodendrocytes. And finally: are the tanycyte-derived cells result from tanycyte cell division or rather (trans)differentiation? It is not clear if the authors tested for these alternatives.

5 On page 7,8 the authors state: "Quantitative analysis confirmed that neural injury increased the number of cycling oligodendrocyte lineage cells and astrocytes, implicating the multipotential differentiation of activated tanycytes into glial cells (Fig. 3h). Nevertheless, neurogenesis in ME was very limited even after neural injury. These results demonstrate that Rax+ tanycytes expand themselves with environmental insult for tissue damage repair." Their data indeed show an injury-induced increase of cell proliferation, however there is no evidence that these new-born EdU+ cells are reporter-positive which would be a required experiment to support the claim that the EdU+ cells indeed originate from tanycytes.

6 Evidence that recombination indeed took place in the Rax-CreER; Igf1r-flx mouse model is lacking

(IHC or in situ for Igflr)

7 Similarly, any direct evidence for activation of the BrafV600E allele is not provided although this might be hard but at least PCR of ME extracts could work or chromosomal FISH...

8 In Fig. 6 the statement: "Rax+ Tanycytes may serve as a cell-of-origin of papillary craniopharyngioma" has no clear experimental support. To make this claim, the authors must show that these new-born EdU+ cells are reporter-positive.

9 In Fig. 8: remove self-renewing tanycytes (or provide evidences - as discussed above)

Reviewer #2 (Remarks to the Author):

The authors investigated the proliferative capacity of tanycytes. They found evidence that tanycytes are a relatively quiescent cell population. In response to mechanical stimulation or cell ablation some degree of proliferation was observed. Moreover, Igf1r seems to be required for the maintenance of tanycyte populations. Most interestingly, a Braf mutation in tanycytes enhanced their proliferation and led to tumor formation in the ME.

1. The manuscript tends to give a biased view on the previous literature that may emphasize the novelty of the current study but is not helpful for readers.
 - a. The introduction does not mention that tanycytes have been reported to proliferate and to have stem cell properties.
 - b. The study by Campbell et al. (2017) did not only focus on neurons but also provided a comprehensive description of tanycytes. In contrast to the present study, the work by Campbell distinguished more than 2 tanycyte subpopulations.
 - c. Scn7a, Col25a1 and Mia are not novel markers for tanycytes as stated in line 96. These genes have been proposed by numerous previous studies as markers of hypothalamic tanycytes.
2. The message of Fig. 1a is unclear. GFAP is no good marker of tanycytes. With this strategy the authors have probably also labeled astrocytes in the ME. Did the authors use this labeling to guide the microdissection?
3. What is the difference between tanycytes 1 and tanycytes 2? To which of the established subpopulations do they correspond? Generally, scRNA seq is a timely method by the conclusions obtained by these experiments is unclear.
4. The authors seem to suggest that Scn7a and Col25a1 expressing cells in the subependymal layer are tanycytes. The consensus is that the cell body of tanycytes is localized in the ventricle wall. Therefore, this rather suggests that some other cells than tanycytes were labeled.
5. Why is the analysis of mitotic genes restricted to Mki67 and Mcm2? Are these the only mitotic genes that were detected? If not, a comprehensive analysis should be performed.
6. To clarify the identity of EdU-positive Sox2-positive cells, Olig2 and Sox2 double staining of EdU-positive cells would be helpful.
7. Where are Olig2-Sox2+EdU+ cells localized after mechanical injury? In the SEZ or EZ?
8. The time course of experiments is often unclear, e.g. Supplementary Fig. 5b. When after the injury were the stainings performed? This information should be included in the figure legends.
9. When ablating tanycytes with the diphtheria approach, it would be necessary to know how many tanycytes were ablated. In this model cell proliferation does not seem to be limited to the tanycytic cell layer but rather involve the whole ME. To conclude the "multipotential differentiation of activated tanycytes" (line 150), the authors should perform tracing studies in Rax-CreERT2 mice.
10. What does "n=3 samples" in the legend mean? Were the samples individual mice or sections of one mouse?
11. Quantification of tanycytes in Fig. 4 should not exclusively rely on Sox2 stains but include other markers. Could Sox2 expression be downregulated in Igf1r-deficient tanycytes?
12. What is the cellular effect of Igf1r? Apparently, tanycytes do not proliferate under basal conditions. Does the absence of Igf1r elicit cell death of tanycytes?
13. A better characterization of the neoplastic properties of tanycytes harboring a Braf mutation is required. Can these cells be transplanted to cause tumors in other animals? Do they proliferate in vitro? Did the authors find metastases?

Reviewer #3 (Remarks to the Author):

I really enjoyed reading this manuscript, it contains a huge amount of work in a series of complementary experiments that really extends our understanding of the biology of tanycytes. I consider the findings to be novel, and add a lot more weight to the concept that these cells are a multipotent stem cell niche. The findings that injury can re-activate them and that they have potential to develop into craniopharyngioma-like tumors is of particular interest. As tanycytes express a very wide range of receptors, perhaps integral to their function of sampling the peripheral circulation and transducing information to the brain, then I think the focus on Igf1r may be somewhat overplayed (see below), but really I have very minor suggestions for revision.

Minor points

Line 30 “Our study offers valuable insights into the properties of tanycytes, which will help us to manipulate tanycyte biology for regulating hypothalamic function and treat clinically relevant tumors.” I agree with the first proposition in that this series of studies does provide valuable insights, but the jump to treatment of tumors is unnecessary exaggeration.

Line 42 “As tanycytes in ME are fundamentally important to maintain systemic homeostasis, their maintenance and homeostasis must be tightly regulated.” I’m not sure that the second point follows or makes much sense. I think it would be sufficient to say “Tanycytes have been implicated in the maintenance of body homeostasis by the hypothalamus, but have also been recently identified as a possible stem cell niche.....”

Line 67 The final paragraph should include a sentence or two to explain what Rax+ tanycytes are, and how they are detected experimentally, referring to the Miranda-Angulo et al 2014 paper here would be helpful. I would also rather see this paragraph outline the objectives/hypothesis of the study rather than a rather bland summary of the outcomes that repeats the abstract.

Line 90/line 164 and later in the manuscript: “the stemness of tanycytes” I’m not familiar with this term, does it mean the potential for tanycytes to maintain multipotency? I suggest it is replaced with a clearer description.

Line 102 “To reveal the identity of dividing cells, we labeled mitotically active cells by ethynyldeoxyuridine (EdU) administration in adult mice” Indicate whether this was by administered in vivo by intracerebroventricular infusion, or systemically eg via intraperitoneal injection as the latter route would label less cells due to BBB permeability issues.

Line 117 What is “leaky labelling”?

Line 124/line 130 and later in the manuscript. I don’t understand what the authors mean when they use the term “homeostasis” to describe a period in time. The term denotes a process of maintaining constant internal conditions, so is always occurring. A different term that is clearly defined should be used.

Line 160 “expression of Igf1r in tanycytes was highly specific, compared to epithelial growth factor receptors” To say ‘highly specific’ is rather a subjective statement based on the figures presented (Fig 4 / Suppl. Fig. 6), and the analyses don’t consider specific splice variants of, for example, FGFRs. Our own work shows high levels of expression of FGFR1c in tanycytes and this signalling input seems to be of biological importance because monoclonal Abs targeting this produce major physiological effects (Samms et al 2015, Current Biology 25:2997-3003), so one might be equally interested in this pathway. I feel the emphasis on IGf1r is more down to the availability of an Igf1r^{f/f} mouse line than more objective considerations.

Line 179 “craniopharyngiomas” = “craniopharyngiomas”

Fran Ebling
University of Nottingham

Response to Reviewer #1

General comments:

The manuscript by Mu and colleagues addresses the proliferative potency of tanycytes in adult mice and their potential role in tumorigenesis. Specifically, they looked into the ability of the median eminence (ME)-lining tanycytes to develop into craniopharyngioma-like tumors upon oncogene activation in the adult brain. By performing single-cell RNAseq analysis of the ME area the authors identified several transcripts that are enriched in tanycytes and confirmed their RNAseq data by single-molecule FISH labeling. Transcriptomic analysis of the cell-cycle mitotic genes revealed that they were predominantly enriched in OPCs but not in the tanycytes. They further confirmed these data by treating mice with mitotic marker EdU and tracing EdU⁺ cells shortly after injections. The oncogene induced proliferation of tanycytes seems to be the remarkable finding of the study, as a single mutation is sufficient to drive proliferation suggesting that tanycytes perhaps represent an exceptionally vulnerable population for tumor induction.

Major weakness (Questions 1-3)

Question 1:

The authors claim that injury or cell ablation induces tanycytic self-renewal and differentiation, and although differentiation was demonstrated by colocalization of reporter (tdTomato) and cell-type markers (S100b or Olig2), self-renewal was never confirmed. To prove the cell-renewal capacity of tanycytes the authors must show triple labeling of their tdTomato reporter with a cell-division marker (i.e. EdU) and a tanycyte marker (i.e. Rax, Scn7a, vimentin, etc).

Reply:

To further confirm the self-renewal capacity of hypothalamic tanycytes after injury, we followed the reviewer's suggestion and performed lineage tracing of Rax⁺ tanycytes using Rax-CreER^{T2}::Ai14 mice. Indeed, a combination of smFISH, immunostaining and chemical labeling revealed a subpopulation of tanycytes showing triple labeling of Scn7a, EdU and tdTomato reporter after neural injury (Fig. 3e and Supplementary Fig. 5b),

suggesting that tanycytes renew themselves by dividing for cell replenishment following injury.

Question 2:

While the oncogene activation results seem exciting, pointing to tanycytes as tumor source, the authors actually do not provide clear direct evidence in support (although they have tools that could easily demonstrate it). Specifically, it is puzzling for the reviewer that the authors did not cross in a fluorescence (e.g. tdTomato) reporter to the Rax-CreER: Braf^{V600E} model. This way, a tanycytic origin of the tumor cells could be easily proven. Making the case for it just based on the absence of Olig2 from newly formed (EdU labelled) Sox2 cells is too indirect.

Reply:

We appreciate the reviewer for this good suggestion. To provide more direct and clear evidence to support that papillary craniopharyngioma (PCP)-like tumor originates from tanycytes, we bred Rax-CreER^{T2}::Braf^{V600E}::Ai14 mice and induced oncogene activation by applying tamoxifen to 2-month-old animals. After 2-month induction, we sacrificed the animals, collected the brain tissues and stained the tumor tissues with EdU and Vimentin antibody. Our results showed that the fast-proliferating tumor cells were basically positive for tdTomato reporter at the floor of the third ventricle (Fig. 6c and Supplementary Fig. 7c), demonstrating the tanycytic origin of PCP-like tumor.

Question 3:

Also at many other occasions the authors distinguish tanycytes from OPCs by double-labeling for Olig2 and Sox2, which is often not very convincing. It is unclear why they did not make more use of the specific tanycytic markers (Rax, Vimentin, etc).

Reply:

Median eminence (ME) is a tiny piece of tissue at the floor of the third ventricle, rendering it to be very fragile during tissue processing. While commercial antibodies targeting against

Rax, Scn7a and Col25a1 did not work well and Vimentin antibody predominantly labeled radial processes of tanycytes but not cell bodies, we had to rely on smFISH to distinguish tanycytes from OPCs for quantification if it is desired to choose more straightforward tanycytic markers. However, heat-based antigen retrieval and ethanol treatment, according to the manufacturer's protocol, frequently made the ME tissue more susceptible to mutilation and caused the data quantification more difficult. Despite these difficulties, we still performed smFISH to quantify the number of tanycytes (dividing or not) in Fig. 1h, 3d and 4d, and found similar results when comparing direct tanycytic markers (Rax or Scn7a) with double-labeling for Olig2 and Sox2. A more gentle smFISH approach is still under development in our lab.

To yield reliable quantitative results in other occasions, we determined to choose the double-staining of Sox2 and Olig2 as an alternative approach to distinguish between tanycytes and OPCs. Based on our single-cell RNAseq data showing that stem cell marker Sox2 labeled both tanycytes and OPCs in ME (Supplementary Fig. 2d and 3h), it is reasonable to speculate that a combination of Sox2 and Olig2 would be able to distinguish tanycytes from OPCs. Taken together, it is our best bet and also convincing enough to quantify the number of tanycytes and OPCs using double-labeling of Sox2 and Olig2. We thank for the reviewer's comment and wish we could make him/her understand the story behind.

The antibodies we tested include: Rax (Rabbit, Proteintech, 25047-1-AP, 1:200), Rax (Rabbit, Abcam, ab23340, 1:200), Rax (Mouse, Santa Cruz, sc-271889, 1:100), Scn7a (Rabbit, Abcam, ab66499, 1:200) and Col25a1 (Rabbit, Biorbyt, orb160512, 1:200).

Other concerns (Questions 4-12)

Question 4:

It would be important to include high-magnification double labeling images for the genes that are claimed as novel tanycytic markers (Scn7a, Col25a1, Mia) with tdTomato.

Reply:

We have added the magnified images showing the colocalization between tancytic markers (Scn7a, Col25a1 and Mia) and tdTomato in the manuscript (Supplementary Fig. 2g). smFISH analyses using probes targeting against Scn7a, Col25a1 and Mia validated the specific expression of these genes in Rax-derived tancytes.

Question 5:

In Suppl. Fig. 5B: S100b/tdTomato double-labeling is somewhat confusing as two arrowheads point to different types of S100b-labeling: the left arrowhead points at a cytoplasmic signal, whereas the right one points at a nuclear signal.

Reply:

Due to the newly added data, the previous Supplementary Fig. 5b in the old version was shifted to Supplementary Fig. 5c. We are grateful to the reviewer for pointing this out. Actually, the right arrowheads point at a fraction of cell body that does not envelop DAPI⁺ nucleus, which could be interpreted as the splitting of a single cell onto two neighboring brain sections. To correct the error, we changed the right arrowheads into dashed circles to distinguish between a complete soma and part of cell body.

Question 6:

Suppl. Fig. 5B indicates DAPI, however the images do not contain a DAPI signal.

Reply:

We have added DAPI signal in Supplementary Fig. 5c (i.e. Supplementary Fig. 5b in the old version).

Question 7:

Suppl. Fig. 5C model is inaccurate as the data do not support it. First, tanycytes have not been demonstrated to self-renew (see above). Secondly, Olig2 labels both OPCs and oligodendrocytes. And finally: are the tanycyte-derived cells result from tanycyte cell division or rather (trans)differentiation? It is not clear if the authors tested for these alternatives.

Reply:

To strengthen the model depicting the potential of hypothalamic tanycytes, we took the reviewer's constructive suggestions, tracked the behavior of Rax⁺ tanycytes following injury and simultaneously labeled mitotic cells in Rax-CreER^{T2}::Ai14 mice. Here we provide point-by-point response to the reviewer's concerns: firstly, we confirmed the self-renew of tanycytes after injury by triple labeling of Scn7a, EdU and tdTomato (Fig. 3e and Supplementary Fig. 5b; See Reply to Question 1); Secondly, we further co-stained tdTomato and EdU with PDGFR α , a more specific OPC marker than Olig2, and found that tanycytes could transform into OPCs by dividing (Fig. 3e), suggesting that injury activates their potential in differentiating into oligodendrocyte lineage; Lastly, we validated a subpopulation of Rax-derived tanycytes showing triple labeling of tdTomato, EdU and Scn7a/PDGFR α /S100 β (markers respectively representing tanycytes, OPCs and astrocytes; Fig. 3e), suggesting that tanycytes divide, at least in part, to self-renew or differentiate into other glial cell types. Nevertheless, we still could not exclude the possibility of trans-differentiation of tanycytes into OPCs and astrocytes following injury, and thereby included this hypothesis in Discussion section (Page 12, Paragraph 2). With the newly added data, we believe that our model has been strongly supported.

Question 8:

On page 7, 8 the authors state: "Quantitative analysis confirmed that neural injury increased the number of cycling oligodendrocyte lineage cells and astrocytes, implicating the

multipotential differentiation of activated tanycytes into glial cells (Fig. 3h). Nevertheless, neurogenesis in ME was very limited even after neural injury. These results demonstrate that Rax⁺ tanycytes expand themselves with environmental insult for tissue damage repair.” Their data indeed show an injury-induced increase of cell proliferation, however there is no evidence that these new-born EdU⁺ cells are reporter-positive which would be a required experiment to support the claim that the EdU⁺ cells indeed originate from tanycytes.

Reply:

In the mechanical injury model, we have provided compelling evidences supporting that new-born EdU⁺ cells are tdTomato reporter-positive and derived from Rax⁺ tanycytes (Fig. 3e; Supplementary Fig. 5b, c). Please find replies to Questions 1 and 7 for detailed description. In the genetically-induced injury model, we further introduced Rosa-DTA (iDTA, JAX#010527) mice to the lab and bred Rax-CreER^{T2}::iDTA::Ai14 mice for genetic injury and lineage tracing experiment. Indeed, we also found the costaining of S100 β or PDGFR α with tdTomato reporter after tamoxifen induction (Supplementary Fig. 5i). Therefore, our results, particularly mechanical injury data, indicate that hypothalamic tanycytes could be activated by injury, self-renew themselves and transform into glial cells.

We also rephrased the related text as follows: Quantitative analysis confirmed that neural injury increased the number of cycling oligodendrocyte lineage cells and astrocytes (Fig. 3i). We also tracked the fate of Rax⁺ tanycytes after genetically-induced injury and found that a subgroup of tanycytes could transform into astrocytes or OPCs (Supplementary Fig. 5i).

Question 9:

Evidence that recombination indeed took place in the Rax-CreER; Igf1r-flx mouse model is lacking (IHC or in situ for Igflr).

Reply:

To validate the CreER-induced deletion of *Igf1r*, we applied four daily tamoxifen injection to *Rax-CreER^{T2}::Igf1r^{f/f}* mice, immunostained brain sections with IGF1R antibody and found a significant reduction of IGF1R signal in ME as compared with control tissues (Supplementary Fig. 6b).

Question 10:

Similarly, any direct evidence for activation of the *Braf^{V600E}* allele is not provided although this might be hard but at least PCR of ME extracts could work or chromosomal FISH.

Reply:

We adopted two approaches to answer this question. We microdissected the tumor tissues, isolated the mRNA, performed reverse transcription to obtain cDNA and amplified the target fragment from cDNA for either restriction digestion or Sanger sequencing. Firstly, we referred to the source paper reporting the generation of *Braf^{V600E}* mouse line ¹, and used *XbaI* restriction enzyme to test the mutation site in *Braf* gene. As compared with control ME extracts, the target DNA fragment derived from *Rax-CreER^{T2}::Braf^{V600E}* mutant mice was susceptible to *XbaI* cut (Supplementary Fig. 7a). Secondly, we sent the target DNA fragment for Sanger sequencing and found a heterozygous double peak at the targeted base pair within *ACAGTGAATCT* loci of mutant *Braf* (Supplementary Fig. 7b), indicating the expression of *Braf^{V600E}* mutant gene in ME tissues.

Question 11:

In Fig. 6 the statement: “*Rax*+ Tanycytes may serve as a cell-of-origin of papillary craniopharyngioma” has no clear experimental support. To make this claim, the authors

must show that these new-born EdU+ cells are reporter-positive.

Reply:

We have shown that the fast-proliferating tumor cells in ME were tdTomato reporter-positive in Rax-CreER^{T2}::Braf^{V600E}::Ai14 mouse model (Fig. 6c and Supplementary Fig. 7c). See Reply to Question 2 for detailed description. In the central zone of neoplastic tissues, EdU⁺ tumor cells were predominantly positive for fluorescent reporter, suggesting that Rax⁺ Tanycytes may serve as cells-of-origin for papillary craniopharyngioma.

Question 12:

In Fig. 8: remove self-renewing tanycytes (or provide evidences - as discussed above)

Reply:

We have followed the reviewer's suggestions and conducted more experiments to provide new evidences on this issue. See replies to Question 1, 7 and 8 for detailed description.

Response to Reviewer #2

General comment

The authors investigated the proliferative capacity of tanycytes. They found evidence that tanycytes are a relatively quiescent cell population. In response to mechanical stimulation or cell ablation some degree of proliferation was observed. Moreover, Igf1r seems to be required for the maintenance of tanycyte populations. Most interestingly, a Braf mutation in tanycytes enhanced their proliferation and led to tumor formation in the ME.

Question 1:

The manuscript tends to give a biased view on the previous literature that may emphasize the novelty of the current study but is not helpful for readers.

a: The introduction does not mention that tanycytes have been reported to proliferate and to have stem cell properties.

b: The study by Campbell et al. (2017) did not only focus on neurons but also provided a comprehensive description of tanycytes. In contrast to the present study, the work by Campbell distinguished more than 2 tanycyte subpopulations.

c: Scn7a, Col25a1 and Mia are not novel markers for tanycytes as stated in line 96. These genes have been proposed by numerous previous studies as markers of hypothalamic tanycytes.

Reply:

We appreciate the reviewer's comments and have revised the text of manuscript according to his/her advice.

a: We have included the results of previous literature reporting the cellular properties of hypothalamic tanycytes in the Introduction section and rephrased the related text by combining the suggestion from Reviewer #3 (Page 3, Paragraph 1). Despite the previous reports showing hypothalamic tanycytes undergo cell division under physiological conditions²⁻⁷, it remains controversial whether tanycytes in ME (β 2 tanycytes) are mitotically active. Firstly, recent studies have shown that there is a substantial enrichment of cell division in ME but most of these proliferating cells could be NG2 glia⁸. Secondly, it has been found that β tanycytes in ME are unable to proliferate to form neurosphere, while α tanycytes above ME are neurosphere-forming⁴. Hence, the disclosure of the regenerative and tumorigenic capacity of ME tanycytes in this study provides several new points of novelty that clearly distinguish it from previous studies. Technically, intraperitoneal injection might be better than intracerebroventricular infusion with regard to EdU/BrdU delivery, which avoids the potential intracranial injury and inflammation. Lastly, the specificity of promoter and leaky expression in previous lineage tracing may

confound the conclusion of hypothalamic neurogenesis. Here we exemplify our own lineage tracing data: we found a subpopulation of labeled neurons in arcuate nucleus using *Rax-CreER^{T2}::Ai14* mice (Supplementary Fig. 4b) but we could not jump to the conclusion that adult neurogenesis occurs in hypothalamus, because our smFISH analysis identified them as leaky labeling. Moreover, given that *Slc1a3* (encoding *Glast*) is highly enriched in astrocytes, we should also be more cautious in the gliogenic potential of α tanycytes while using *Glast-CreER^{T2}* transgenic mice to investigate their potential^{9,10}.

b: Campbell and his colleagues indeed identified more than 2 subpopulations of tanycytes. The clustering of single-cell RNAseq data frequently relies on the number of detected cells, the definition of resolution parameter and the goal of research. In one of our ongoing studies, we also revealed more than 2 subpopulations of tanycytes arising from *Tbx3*-positive neural progenitors during development (Fig. R1, unpublished data). However, here we aim to reveal the regenerative and tumorigenic potential of tanycytes in ME but not the taxonomy of tanycytes lining the third ventricle. Hence, we only microdissected ME region (containing $\beta 2$ tanycytes) for single-cell RNAseq and were devoted to uncover the general markers and properties of ME tanycytes using our single-cell dataset, but did not subcluster the tanycytes at higher resolution.

Fig. R1 Tbx3-derived tanycytes in hypothalamus. a, tSNE clustering of tanycytes arising from *Tbx3*⁺

neural progenitors. We applied a single dose of tamoxifen at E9 in Tbx3-CreER^{T2};Ai14 mice and sorted the tdTomato⁺ cells in hypothalamus for single-cell RNAseq. **b**, Dot plot showing the specific markers for different tanycyte clusters. **c**, Expression of tdTomato in tanycytes lining the third ventricle (left) and the marker gene expression shown by in situ hybridization of coronal brain sections (Allan Mouse Brain Atlas).

c: We carefully reviewed the literature and agreed with the reviewer's comments in that Scn7a and Col25a1 have been identified as markers for ME tanycytes (β 2 subtype) in the study by Campbell et al. (2017). However, I think we confirmed these markers by smFISH for the first time (Fig. 1d-f and Supplementary 2g). In the manuscript, we removed "novel" and revised the text to be more rigorous. We appreciate all of these reviewer's comments and reminders.

Question 2:

The message of Fig. 1a is unclear. GFAP is no good marker of tanycytes. With this strategy the authors have probably also labeled astrocytes in the ME. Did the authors use this labeling to guide the microdissection?

Reply:

Yes, we used GFAP-based labeling to guide the clearcut microdissection of ME but did not intend to use GFAP as a marker for tanycytes.

Question 3:

What is the difference between tanycytes 1 and tanycytes 2? To which of the established subpopulations do they correspond? Generally, scRNAseq is a timely method by the conclusions obtained by these experiments is unclear.

Reply:

In the revised manuscript, we upgraded single-cell analysis software from Seurat 2.0 to 3.0, re-analyzed our dataset and identified only one cluster of tanycytes using the same parameters (Fig. 1b, c; Supplementary Fig. 2a, d). Despite that *Ces1d* and *Apoc3* remained to be specifically expressed in cluster 1 (c1), we identified c1 as pars tuberalis cells (PTCs) due to their specific expression of *Tshb* and *Cck* (Supplementary Fig. 2b) ¹¹.

The single-cell database in the manuscript is relatively small and a larger database covering different age, gender, mouse lines and species is cumulatively expanding in our lab. Nevertheless, our current single-cell analysis still revealed that: 1) tanycytes in ME are not mitotic active under physiological conditions (Supplementary Fig. 3a, b); 2) *Sox2* in combination with *Olig2* distinguishes tanycytes from OPCs in ME (Supplementary Fig. 3h); and 3) a specific set of receptor tyrosine kinases (RTKs) are expressed in tanycytes (Supplementary Fig. 6a). We understand the reviewer's concerns, but here we would like to emphasize that single-cell taxonomy of tanycytes is not our key point in the manuscript. However, the cumulating single-cell database of tanycytes has been helping us gain deep insight into the molecular and cellular properties of hypothalamic tanycytes.

Question 4:

The authors seem to suggest that *Scn7a* and *Col25a1* expressing cells in the subependymal layer are tanycytes. The consensus is that the cell body of tanycytes is localized in the ventricle wall. Therefore, this rather suggests that some other cells than tanycytes were labeled.

Reply:

We agree with reviewer's opinion that α tanycytes are located in the ependymal layer lining the third ventricle, but β 2 tanycytes may exist in both ependymal and subependymal layers within ME. Here we only found a single subpopulation of cells specifically expressing *Rax*, *Scn7a*, *Col25a1* and *Mia* (Fig. 1c and Supplementary Fig. 2a). Both lineage tracing using *Rax-CreER*^{T2} mice and smFISH analysis supported the existence of a large amount of

tanycytes (Rax⁺, Scn7a⁺ and Col25a⁺ cells) residing in subependymal zone (SEZ; Fig. 1d, 1e, 2a, 2b and Supplementary Fig. 3d, 4b). If there existed another subpopulation of cells diversifying from tanycytes but expressing Rax, Scn7a, Col25a1 and Mia, we should be able to detect them in our single-cell analysis and identify their molecular profile. Therefore, these data suggest that tanycytes may also reside in SEZ within ME, a specialized niche.

Besides, we would like to take radial glia cells (RGCs) in outer subventricular zone as an example to support us. There was a consensus that the cell body of RGCs were exclusively located in ventricular zone of developing cerebral cortex before 2010, but Arnold Kriegstein's lab reported that neurogenic RGCs were also distributed in outer subventricular zone and proliferated to increase neuronal numbers ¹².

Question 5:

Why is the analysis of mitotic genes restricted to Mki67 and Mcm2? Are these the only mitotic genes that were detected? If not, a comprehensive analysis should be performed.

Reply:

Mki67 and Mcm2 are among the most well-known mitotic genes. To provide a more comprehensive view of the mitotic activity among diverse clusters of cells, we analyzed the average expression of 97 mitotic genes (See Methods and Materials – Single-cell RNAseq analysis) in our single-cell dataset and found a similar result when comparing with the tSNE plot covering only Mki67 and Mcm2 (Supplementary Fig. 3a, b).

Question 6:

To clarify the identity of EdU-positive Sox2-positive cells, Olig2 and Sox2 double staining of EdU-positive cells would be helpful.

Reply:

We have performed double immunostaining of Sox2 and Olig2 of EdU-positive cells and updated the quantitative data in Fig. 2e, 2i and 5e. For many other occasions (e.g. Fig. 3d, 3h, 4d, 4g and 6f), we had already co-stained the brain sections with EdU, Sox2 and Olig2 to distinguish the identity of EdU-positive cells.

Question 7:

Where are Olig2-Sox2+EdU+ cells localized after mechanical injury? In the SEZ or EZ?

Reply:

In the mechanical injury model, the tanocytes derived from EZ frequently relocated to SEZ by the mechanical force. We did observe the distribution of Olig2-Sox2+EdU+ cells in both EZ and SEZ after injury (Fig. 3b, c), but it was difficult to precisely quantify the number/percentage of these cells positioned in either EZ or SEZ because the morphology of ME was frequently partially impaired by mechanical injury.

Question 8:

The time course of experiments is often unclear, e.g. Supplementary Fig. 5b. When after the injury were the stainings performed? This Information should be included in the figure legends.

Reply:

We apologize for the confusing information and have updated the figure legends (i.e. Supplementary Fig. 5c in current version). To investigate the potential of Rax+ tanocytes following injury, we applied a single dose of tamoxifen to label tanocytes in Rax-CreER^{T2}::Ai14 mice, subjected the animals to mechanical injury at 7 days post induction

(dpi) and injected EdU daily for three consecutive days at 1 day after mechanical injury, followed by tissue collection for analysis. Briefly, the staining was performed at 4 days after injury.

Question 9:

When ablating tanycytes with the diphtheria approach, it would be necessary to know how many tanycytes were ablated. In this model cell proliferation does not seem to be limited to the tanycytic cell layer but rather involve the whole ME. To conclude the “multipotential differentiation of activated tanycytes” (line 150), the authors should perform tracing studies in Rax-CreERT2 mice.

Reply:

We appreciate the reviewer’s insightful suggestion. To address the reviewer’s concerns, we firstly quantified the number of GFP-labeled cells before and after genetic ablation and found that approximately 100 cells/brain section were genetically ablated (Supplementary Fig. 5g). Indeed, the cell proliferation occurs in ependymal and subependymal layers in both neural injury models (Fig. 3b, c and Supplementary Fig. 5h). In the mechanical injury model, we have performed lineage tracing using Rax-CreER^{T2}::Ai14 mice and provided more evidences supporting that Rax-derived tanycytes could differentiate into astrocytes (EdU⁺tdTomato⁺S100 β ⁺) or OPCs (EdU⁺tdTomato⁺PDGFR α ⁺) at 4 days after injury (Fig. 3e and Supplementary Fig. 5c). In the genetic injury model, we bred Rax-CreER^{T2}::iDTA::Ai14 mice and also found the costaining of S100 β or PDGFR α with tdTomato reporter after tamoxifen-induced injury, suggesting the multipotential differentiation of tanycytes (Supplementary Fig. 5i).

Question 10:

What does “n=3 samples” in the legend mean? Were the samples individual mice or

sections of one mouse?

Reply:

The “n” values in Fig. 1h, 2c and 2e represent the number of animals to quantify the percentage of cells, whereas the values in Fig. 3d, 3h, 3i, 4d, 4g, 5c, 5e, 6b, 6f, Supplementary Fig. 5a, 5g, 5h and 6e represent the number of sections to analyze the number of cells per section (See Methods and Materials – *Cell quantification*). In the manuscript, at least three animals and five brain sections from each animal were used for quantitative analyses. To clarify this, we have amended all of related figure legends.

Question 11:

Quantification of tanycytes in Fig. 4 should not exclusively rely on Sox2 stains but include other markers. Could Sox2 expression be downregulated in Igf1r-deficient tanycytes?

Reply:

To corroborate the cellular phenotype of Igf1r-deficient mice, we further performed smFISH on brain sections covering ME and quantified the number of Scn7a⁺ tanycytes in SEZ (Fig. 4d and Supplementary Fig. 6c). Our results showed that the amount of Scn7a⁺ cells was significantly declined in Igf1r-defective brains, confirming previous results that relied on double immunostaining of Sox2 and Olig2.

While Sox2 staining signals were strikingly compromised within ME of Igf1r-deficient mice (Fig. 4c, d), it is interesting to investigate whether Sox2 expression level is downregulated by loss of IGF1R. We thank the reviewer for his/her valuable advice. Given that β tanycytes within ME are not neurosphere-forming⁴, we determined to culture mouse neural stem cells *in vitro* and treated the cells with OSI-906 to inhibit IGF1R signaling for 48 hrs. Immunoblotting analysis showed that the treatment subtly reduced the expression level of Sox2 (Supplementary Fig. 6f-h). Collectively, our data suggest that Igf1r signaling is essential for maintaining tanycyte population and their stemness property.

Question 12:

What is the cellular effect of Igf1r? Apparently, tanycytes do not proliferate under basal conditions. Does the absence of Igf1r elicit cell death of tanycytes?

Reply:

The reviewer's insight is greatly appreciated. To investigate the role of IGF1R signaling in maintaining the viability of tanycytes, we performed TUNEL assay on the brain sections of Rax-CreER^{T2}::Igf1r^{f/f} mice, and found that the number of apoptotic cells was significantly increased after the deletion of Igf1r in tanycytes. In conjunction with the Reply to Question 11, our results suggest that IGF1R signaling plays an essential role in maintaining the stemness of tanycytes by preserving the expression of stemness-related proteins and preventing the apoptosis of tanycytes.

Question 13:

A better characterization of the neoplastic properties of tanycytes harboring a Braf mutation is required. Can these cells be transplanted to cause tumors in other animals? Do they proliferate in vitro? Did the authors find metastases?

Reply:

We are grateful to the reviewer for these great suggestions. To characterize the papillary craniopharyngioma (PCP)-like tumor, we took the reviewer's advices and performed xenograft transplantation experiment, *in vitro* culture of tumor cells and histological analysis of multiple organs in the tumor-carrying mice. Firstly, we found that the tumor cells derived from induced Rax-CreER^{T2}::Braf^{V600E} mice did not cause neoplasm in nude mice at 7 and 15 days post inoculation, whereas the transplantation of melanoma cells induced obvious subcutaneous tumors (Supplementary Fig. 8a). Secondly, we did not

observe significant metastasis of tumor cells arising from tanycytes in multiple organs even at 6 months post induction (mpi), including other brain regions, liver, spleen, lung and kidney (Supplementary Fig. 8b, c). Lastly, we cultured the tumor cells *in vitro* but found that these cells failed to survive the second passage, whereas melanoma cells proliferate, expand and maintain themselves through as many as five passages (Supplementary Fig. 8d). Together, our data support that the PCP-like tumor is generally benign neoplasm, consistent with the pathological features of PCP in the clinic¹³⁻¹⁸. However, the patients tend to exhibit severe, debilitating symptoms due to the specific anatomic location of PCP.

Response to Reviewer #3

General comment

I really enjoyed reading this manuscript, it contains a huge amount of work in a series of complementary experiments that really extends our understanding of the biology of tanycytes. I consider the findings to be novel, and add a lot more weight to the concept that these cells are a multipotent stem cell niche. The findings that injury can re-activate them and that they have potential to develop into craniopharyngioma-like tumors is of particular interest. As tanycytes express a very wide range of receptors, perhaps integral to their function of sampling the peripheral circulation and transducing information to the brain, then I think the focus on Igf1r may be somewhat overplayed (see below), but really I have very minor suggestions for revision.

Reply

We appreciate the reviewer's very positive comments on our work.

Minor points

Question1:

Line 30 "Our study offers valuable insights into the properties of tanycytes, which will help

us to manipulate tanycyte biology for regulating hypothalamic function and treat clinically relevant tumors.” I agree with the first proposition in that this series of studies does provide valuable insights, but the jump to treatment of tumors is unnecessary exaggeration.

Reply

We thank the reviewer for pointing this out and have toned down the statement by changing the sentence into “Our study offers valuable insights into the properties of tanycytes, which will help us to manipulate tanycyte biology for regulating hypothalamic function and investigate the pathogenesis of clinically relevant tumors” (Page 2).

Question2:

Line 42 “As tanycytes in ME are fundamentally important to maintain systemic homeostasis, their maintenance and homeostasis must be tightly regulated.” I’m not sure that the second point follows or makes much sense. I think it would be sufficient to say “Tanycytes have been implicated in the maintenance of body homeostasis by the hypothalamus, but have also been recently identified as a possible stem cell niche.....”

Reply

We combined the suggestion of Reviewer #2 and #3, and revised the related text as follows:

Tanycytes in ME have been implicated in the maintenance of hypothalamus-mediated body homeostasis, but have also been recently identified as a possibly key component of stem cell niche due to their prominent expression of neural progenitor markers and their potency of proliferation and differentiation (Page 3, Paragraph 1).

Question 3:

Line 67 The final paragraph should include a sentence or two to explain what Rax+

tanycytes are, and how they are detected experimentally, referring to the Miranda-Angulo et al 2014 paper here would be helpful. I would also rather see this paragraph outline the objectives/hypothesis of the study rather than a rather bland summary of the outcomes that repeats the abstract.

Reply

We appreciate the reviewer's great advice. To define Rax⁺ tanycytes and outline the objective/hypothesis of our study, we rephrased the last paragraph in Introduction section as follows:

Retina and anterior neural fold homeobox transcription factor (Rax) is selectively expressed in hypothalamic tanycytes, especially those at the ventral part of the third ventricle¹⁹. Here we aim to deconstruct the regenerative and tumorigenic potential of Rax⁺ tanycytes and speculate that whether tanycytes in ME serve as cells-of-origin for craniopharyngioma. We thereby manipulated tanycyte biology and performed lineage tracing using Rax-CreER^{T2} knockin mice, and revealed that Rax⁺ tanycytes responded to neural injury for regeneration and contributed to tumorigenesis upon Braf oncogene activation.

Question 4:

Line 90/line 164 and later in the manuscript: "the stemness of tanycytes" I'm not familiar with this term, does it mean the potential for tanycytes to maintain multipotency? I suggest it is replaced with a clearer description.

Reply

The reviewer's understanding of this term is basically correct. To be more precise, stemness refers to the common molecular features underlying the core stem cell properties of self-renewal and the generation of differentiated progeny. For Line 90 in the old version, we compared the stemness-related genes among tanycytes, NSCs and ependymal cells,

which reflect the molecular features of neural stem/progenitor cells. Although we could not find a better word in Line 90, we replaced the “stemness” with “number” to facilitate the understanding by readers in old-version Line 164 (Page 9, Paragraph 1).

Question 5:

Line 102 “To reveal the identity of dividing cells, we labeled mitotically active cells by ethynyldeoxyuridine (EdU) administration in adult mice” Indicate whether this was by administered in vivo by intracerebroventricular infusion, or systemically eg via intraperitoneal injection as the latter route would label less cells due to BBB permeability issues.

Reply

To clearly describe the delivery route of EdU, we have updated the text by adding “intraperitoneal” in the sentence (Page 6, Paragraph 2) and figure legend. We labeled the mitotic cells in median eminence (ME) by intraperitoneally injecting EdU into animals to avoid the potential intracranial injury and inflammation. As one of the seven brain areas devoid of a blood-brain barrier (BBB), ME is a circumventricular organ with extraordinary permeability. Therefore, we don’t think the intraperitoneal injection of EdU would reduce the efficiency of labeling in this particular region, but intracerebroventricular infusion of EdU continuously labels the mitotic cells with higher dose and longer duration.

Question 6:

Line 117 What is “leaky labelling”?

Reply:

CreER^{T2} mouse lines are commonly used in combination with reporter genes for lineage

tracing and mosaic analysis. The expression of reporter genes in labeled cells requires the tamoxifen-induced translocation of Cre recombinase into nucleus and subsequent removal of STOP cassette before reporter genes. “Leaky labeling” refers to the nonspecific recombination and labeling of cells without the expression of Cre driver genes (e.g. Rax in the manuscript). In Supplementary Fig 4b, we found a few neurons in arcuate nucleus were labeled by fluorescent reporter but lacked expression of Scn7a or Rax at 1 day post tamoxifen injection, suggesting the leaky labeling of Rax-CreER^{T2}::Ai14.

Question 7:

Line 124/line 130 and later in the manuscript. I don’t understand what the authors mean when they use the term “homeostasis” to describe a period in time. The term denotes a process of maintaining constant internal conditions, so is always occurring. A different term that is clearly defined should be used.

Reply:

We have changed the term “homeostasis” into “under physiological conditions” in old-version Line 124 (Page 7, Paragraph 1), and removed the “homeostasis” in old-version Line 130 without changing the information we want to convey (Page 7, Paragraph 2).

Question 8:

Line 160 “expression of Igf1r in tanycytes was highly specific, compared to epithelial growth factor receptors” To say ‘highly specific’ is rather a subjective statement based on the figures presented (Fig 4 / Suppl. Fig. 6), and the analyses don’t consider specific splice variants of, for example, FGFRs. Our own work shows high levels of expression of FGFR1c in tanycytes and this signalling input seems to be of biological importance because monoclonal Abs targeting this produce major physiological effects (Samms et al 2015, Current Biology 25:2997-3003), so one might be equally interested in this pathway. I feel

the emphasis on IGF1r is more down to the availability of an IGF1r^{f/f} mouse line than more objective considerations.

Reply:

We thank the reviewer for his nice advice and sharing the research in his lab. Firstly, we agree with the reviewer in defining the expression of IGF1r in tanycytes as “relatively specific” rather than “highly specific”. Secondly, given that 10×Genomics-based single-cell RNAseq was not designed to distinguish between different splice variants of a gene, we are not able to perform the analysis of splicing isoforms of FGFRs. Lastly, it would be very interesting to investigate the role of FGFR1 in maintaining tanycytes in the near future.

Question 9:

Line 179 “carniopharyngiomas” = “craniopharyngiomas”

Reply:

We apologize for this glitch and have corrected this typo.

Reference

1. Dankort, D. *et al.* A new mouse model to explore the initiation, progression, and therapy of BRAFV600E-induced lung tumors. *Genes Dev* **21**, 379-384 (2007).
2. Goodman, T. & Hajihosseini, M.K. Hypothalamic tanycytes-masters and servants of metabolic, neuroendocrine, and neurogenic functions. *Front Neurosci* **9**, 387 (2015).
3. Lee, D.A. *et al.* Tanycytes of the hypothalamic median eminence form a diet-responsive neurogenic niche. *Nat Neurosci* **15**, 700-702 (2012).
4. Robins, S.C. *et al.* alpha-Tanycytes of the adult hypothalamic third ventricle include distinct populations of FGF-responsive neural progenitors. *Nat Commun* **4**, 2049 (2013).
5. Rodriguez, E.M. *et al.* Hypothalamic tanycytes: a key component of brain-endocrine interaction. *Int Rev Cytol* **247**, 89-164 (2005).
6. Haan, N. *et al.* Fgf10-expressing tanycytes add new neurons to the appetite/energy-balance regulating centers of the postnatal and adult hypothalamus. *J Neurosci* **33**, 6170-6180 (2013).
7. Kokoeva, M.V., Yin, H. & Flier, J.S. Neurogenesis in the hypothalamus of adult mice: potential role in energy balance. *Science* **310**, 679-683 (2005).
8. Djogo, T. *et al.* Adult NG2-Glia Are Required for Median Eminence-Mediated Leptin Sensing and Body Weight Control. *Cell Metab* **23**, 797-810 (2016).
9. Mori, T. *et al.* Inducible gene deletion in astroglia and radial glia--a valuable tool for functional and lineage analysis. *Glia* **54**, 21-34 (2006).
10. Srinivasan, R. *et al.* New Transgenic Mouse Lines for Selectively Targeting Astrocytes and Studying Calcium Signals in Astrocyte Processes In Situ and In Vivo. *Neuron* **92**, 1181-1195 (2016).
11. Campbell, J.N. *et al.* A molecular census of arcuate hypothalamus and median eminence cell types. *Nat Neurosci* **20**, 484-496 (2017).
12. Hansen, D.V., Lui, J.H., Parker, P.R. & Kriegstein, A.R. Neurogenic radial glia in the outer subventricular zone of human neocortex. *Nature* **464**, 554-561 (2010).
13. Muller, H.L. Craniopharyngioma. *Endocr Rev* **35**, 513-543 (2014).

14. Muller, H.L., Merchant, T.E., Warmuth-Metz, M., Martinez-Barbera, J.P. & Puget, S. Craniopharyngioma. *Nat Rev Dis Primers* **5**, 75 (2019).
15. Larkin, S.J. & Ansorge, O. Pathology and pathogenesis of craniopharyngiomas. *Pituitary* **16**, 9-17 (2013).
16. Brastianos, P.K. *et al.* Exome sequencing identifies BRAF mutations in papillary craniopharyngiomas. *Nat Genet* **46**, 161-165 (2014).
17. Karavitaki, N. & Wass, J.A. Craniopharyngiomas. *Endocrinol Metab Clin North Am* **37**, 173-193, ix-x (2008).
18. Fernandez-Miranda, J.C. *et al.* Craniopharyngioma: a pathologic, clinical, and surgical review. *Head Neck* **34**, 1036-1044 (2012).
19. Miranda-Angulo, A.L., Byerly, M.S., Mesa, J., Wang, H. & Blackshaw, S. Rax regulates hypothalamic tanycyte differentiation and barrier function in mice. *J Comp Neurol* **522**, 876-899 (2014).

Reviewers' comments:

Reviewer #1 (Remarks to the Author):

The authors addressed most comments to my satisfaction. The only remaining concern is the statement that tanycytes differentiate into OPCs and astrocytes upon neural injury. The new images presented in the manuscript that lead to this conclusion are of poor quality and are not convincing (unlike to other images in the study). Specifically, the green (Scn7a) channel of Fig. 3e shows a rather amorph staining that precludes discerning individual Scn7a cells. The same applies for the tdTomato signal. The low quality of this critical images is a bit puzzling as for other figures, tdTomato labeling shows excellent single soma/process resolution (for instance in Suppl. Fig 4b individual tdTomato+ cells are easily identifiable).

Even less convincing is the middle panel of Fig. 3e: it is absolutely impossible to identify individual tdTomato+ cells (somas/processes) that correspond to the two highlighted EdU+ cells.

Same applies to Suppl. Figure 5c: again, it is very hard to match S100b+ and Olig2+ cells with tdTomato+ cells. Additionally, the DAPI signal here is extremely overexposed which precludes single cell identification.

Also, these cells are found 4 days post-EdU injection, which seems an extremely short time interval for differentiation to take place.

This shortcoming is also evident in the Figures dealing with DT-induced cell differentiation in Suppl. Figure 5i:

- Left panel: all three channels appear blurry precluding any clear conclusion.

- Right panel: what's been identified as a PDGFRa+ cell lacks the polydendrocytic morphology of OPCs (e.g. see OPC in cartoon of Suppl. Fig. 5d). It rather appears very tanycytic not the least due to the fact that the soma is flush with the third ventricle, a behavior typically not seen with cells other than tanycytes or ependymocytes.

Note, that it has been reported that dsRed derivatives (tdTomato being one of them) can be photoconverted (also known as "greening" see for instance Kremers GJ, et al. Nat Methods. 2009).

Thus perhaps the green signal in the right-hand panel is due to 'greening' of the tdTomato chromophore, which would explain the striking tanycytic morphology of the cell shown.

Due to these shortcomings and consequential uncertainties regarding data interpretation, the sections dealing with the mechanical/toxin-induced differentiation of tanycytes into OPCs and astrocytes should be removed from the manuscript entirely unless much more robust data is provided.

Reviewer #2 (Remarks to the Author):

The authors have addressed most of the reviewers' concerns. However, still a few minor issues are open.

1. How many mice were transplanted with tumor cells? In how many Rax-CreERT2::BrafV600E mice were tumors observed? Do all animals develop tumors?
2. During the revision the authors have corrected the annotation of cell clusters leading to only one tanycyte cell cluster (cluster 2). Please correct the red background label in Fig. 4a accordingly.
3. In Supplemental Fig. 6d and e, it would be necessary to mention which tissue has been stained.

Reviewer #3 (Remarks to the Author):

This manuscript has been very thoroughly revised and improved in response to the three reviewers'

initial comments. A significant amount of additional experimental work has been carried out that addresses the majority of previously raised concerns.

Reviewer #4 (Remarks to the Author):

I really enjoyed reading the manuscript and appreciated the conceptual advance that the authors have made. The authors revealed the physiological characteristics of tanycytes in ME, which are largely quiescent but quickly self-renewed and regenerated upon neural injury, through Igf1r signaling. Moreover, they introduced a Braf-mutation in ME tanycytes and found the potential tumorigenic capacity of tanycytes and the subsequent formation of papillary craniopharyngioma (PCP)-like tumor, providing a novel pathogenic mechanism underlying the development of PCP in patients. I think this study is highly innovative and potentially opens up a completely new field for the researchers and neurosurgeons investigating the pathogenic mechanism of craniopharyngioma.

Question 1:

As a traditional assumption, craniopharyngiomas (CPs) originates from the remnants of the craniopharyngeal duct epithelium (also known as Rathke's pouch) and CPs may develop at any anatomic position along the pituitary–hypothalamic axis, from the sella turcica to the third ventricle of the brain. Tumor derived from tanycytes in ME could elucidate CPs located in infundibulum and tuber cinereum regions, I am wondering how about the CPs located in low part stalk and within sella.

Question 2:

Many tumors harboring Braf mutation are malignant, rather than benign tumors, such as melanoma, colorectal cancer, breast carcinoma and glioblastoma. This study had induced a PCP-like tumor in Rax+ tanycytes, but whether the pituitary cells harboring Braf mutation display the same tumorigenic potential and develop into adamantinomatous craniopharyngioma (ACP) or PCP.

Question 3

CPs was assumed to be derived from squamous epithelial cells by clinical researchers, instead of neuroepithelial cells, despite the evidence has long been lacking. The authors would better explain and discuss the other potential origins of CPs.

Question 4

Although the authors showed that PCP-like tumors expressed tanycyte markers such as Ptpnz1, Fabp7 and Crym, does the transcriptome of PCP-like tumors display any similarity with squamous epithelial cells (e.g. esophageal or oropharyngeal squamous epithelial cells)?

Response to Reviewer #1

Question 1:

The authors addressed most comments to my satisfaction. The only remaining concern is the statement that tanycytes differentiate into OPCs and astrocytes upon neural injury. The new images presented in the manuscript that lead to this conclusion are of poor quality and are not convincing (unlike to other images in the study). Specifically, the green (Scn7a) channel of Fig. 3e shows a rather amorph staining that precludes discerning individual Scn7a cells. The same applies for the tdTomato signal. The low quality of this critical images is a bit puzzling as for other figures, tdTomato labeling shows excellent single soma/process resolution (for instance in Suppl. Fig 4b individual tdTomato+ cells are easily identifiable).

Even less convincing is the middle panel of Fig. 3e: it is absolutely impossible to identify individual tdTomato+ cells (somas/processes) that correspond to the two highlighted EdU+ cells.

Same applies to Suppl. Figure 5c: again, it is very hard to match S100b+ and Olig2+ cells with tdTomato+ cells. Additionally, the DAPI signal here is extremely overexposed which precludes single cell identification.

Reply:

We appreciate the reviewer's comments and apologize for the low-resolution images provided in Fig. 3e and S5c in our manuscript. To improve the quality of these images, we tried our best to perform additional experiments, collected another batch of data and replaced the previous images (See new Fig. 3e and S5c). Our data showed that Rax⁺ tanycytes predominantly underwent self-renewal to replenish themselves after injury and a minority of them differentiated into OPC or astrocytes. Hopefully, our new images are satisfactory.

Technically, mechanical injury frequently disrupted the anatomic organization of hypothalamic ME and heat-based antigen retrieval impaired the brightness and intactness of tdTomato signal to a certain extent, even after the antibody staining. These technical issues have long impeded us from collecting super-perfect

data. Given these issues, we have also toned down the statement that tanycytes differentiate into OPCs and astrocytes upon neural injury in the text and models. We thank the reviewer for understanding.

Question 2:

Also, these cells are found 4 days post-EdU injection, which seems an extremely short time interval for differentiation to take place.

Reply:

We thank the reviewer's great concern. We determined to sacrifice the animals for analysis at 4 days post-EdU injection for the following reasons: 1) it has previously reported that adult neural stem cells undergo differentiation at 2 days after genetic labeling and display a robust transcriptomic change at 2 days after ischemic injury (1); 2) ependymal cells lining the lateral ventricles have also been found to differentiate into astrocytes at 3 days after stroke (2, 3); 3) our previous preliminary experiment suggested that the glial scar formed at 2 weeks after mechanical injury may confound our analysis of the potential of hypothalamic tanycytes. However, in the genetic injury experiment, we analyzed the fate of tanycytes at 10 days post the first tamoxifen administration. Despite these reasons, we agree with the reviewer that the tanycytes may take longer time for differentiation upon injury.

Question 3:

This shortcoming is also evident in the Figures dealing with DT-induced cell differentiation in Suppl. Figure 5i:

- Left panel: all three channels appear blurry precluding any clear conclusion.
- Right panel: what's been identified as a PDGFRa+ cell lacks the polydendrocytic morphology of OPCs (e.g. see OPC in cartoon of Suppl. Fig. 5d). It rather appears very tanycytic not the least due to the fact that the soma is flush with the third ventricle, a behavior typically not seen with cells other than tanycytes or ependymocytes. Note, that it has been reported that dsRed derivatives (tdTomato being one of them) can be photoconverted (also known as "greening" see for instance Kremers GJ, et al. Nat

Methods. 2009). Thus perhaps the green signal in the right-hand panel is due to ‘greening’ of the tdTomato chromophore, which would explain the striking tanycytic morphology of the cell shown.

Reply:

To overcome these shortcomings, we have performed more experiments and replaced the low-quality images with new data (new Fig. S5i). Again, we have to stress that the heat-based antigen retrieval seems to impair the brightness and intactness of tdTomato signal in our hand, especially the cells residing in ependymal zone (EZ). However, our data still implicated that a minority of tanycytes differentiated into OPCs or astrocytes after injury. We highly appreciate the reviewer’s intent to improve the quality of our manuscript.

Question 4:

Due to these shortcomings and consequential uncertainties regarding data interpretation, the sections dealing with the mechanical/toxin-induced differentiation of tanycytes into OPCs and astrocytes should be removed from the manuscript entirely unless much more robust data is provided.

Reply:

As aforementioned, we have replaced all of the previous low-quality images with better data. Given the reviewer’s concern and that only a small number of tanycytes were found to differentiate into OPCs or astrocytes, we have toned down the conclusion in our text. Again, we thank all of the reviewer’s suggestions and concerns.

Reviewer #2

Question 1:

How many mice were transplanted with tumor cells? In how many Rax-CreERT2::iBraf^{V600E} mice were tumors observed? Do all animals develop tumors?

Reply:

In the transplantation assay, we microdissected the tumors from 10 induced Rax-CreER^{T2}::iBraf^{V600E} mice and transplanted these craniopharyngioma-like tumor cells into 3 nude mice, which did not develop neoplasm at 15 days after transplantation. Correspondingly, we dissected the neoplasm from the skin of 3 induced Tyr-CreER^{T2}::iBraf^{V600E}::Pten^{Flox/Flox} mice and performed xenograft transplantation of melanoma cells into 3 nude mice, which caused a significant subcutaneous neoplasm at 7 and 15 days after transplantation.

Although we only used 4 mice for quantification and 3 mice for bulk RNAseq in the manuscript, we have sacrificed at least 30 induced Rax-CreER^{T2}::iBraf^{V600E} and found that all of them developed neoplasm with relatively variable size.

Question 2:

During the revision the authors have corrected the annotation of cell clusters leading to only one tanycyte cell cluster (cluster 2). Please correct the red background label in Fig. 4a accordingly.

Reply:

We thank the reviewer for pointing this out. We apologize for the glitch and have corrected the red background label in Fig. 4a.

Question 3:

In Supplemental Fig. 6d and e, it would be necessary to mention which tissue has been

stained.

Reply

We apologize for the incomplete information and have updated the figure legend. In Supplemental Fig. 6d and e, median eminence (ME) was targeted for TUNEL (terminal deoxynucleotidyl transferase dUTP nick end labeling) staining.

Response to Reviewer #4

General comment:

I really enjoyed reading the manuscript and appreciated the conceptual advance that the authors have made. The authors revealed the physiological characteristics of tanycytes in ME, which are largely quiescent but quickly self-renewed and regenerated upon neural injury, through Igf1r signaling. Moreover, they introduced a Braf-mutation in ME tanycytes and found the potential tumorigenic capacity of tanycytes and the subsequent formation of papillary craniopharyngioma (PCP)-like tumor, providing a novel pathogenic mechanism underlying the development of PCP in patients. I think this study is highly innovative and potentially opens up a completely new field for the researchers and neurosurgeons investigating the pathogenic mechanism of craniopharyngioma.

Reply:

We appreciate the reviewer's positive comments on our work.

Question 1:

As a traditional assumption, craniopharyngiomas (CPs) originates from the remnants of the craniopharyngeal duct epithelium (also known as Rathke's pouch) and CPs may develop at any anatomic position along the pituitary–hypothalamic axis, from the sella turcica to the third ventricle of the brain. Tumor derived from tanycytes in ME could elucidate CPs located in infundibulum and tuber cinereum regions, I am wondering how about the CPs located in low part stalk and within sella.

Reply:

We appreciate the reviewer for raising this issue. Indeed, clinical observations have suggested that CPs may have different origins along the infundibular stalk (4). While our results suggest that tanycytes in the hypothalamic ME serve as the cells-of-origin for CP located in upper part of infundibular stalk, infundibulum or tuber cinereum

regions, recent studies have demonstrated that pituitary progenitors derived from Rathke's pouch could contribute to the formation of CP in lower part of infundibular stalk or within sella (5-7). Moreover, the tumors originating from embryonic remnants of the Rathke's pouch epithelium are more likely paediatric ACP rather than adult-onset PCP.

Question 2:

Many tumors harboring Braf mutation are malignant, rather than benign tumors, such as melanoma, colorectal cancer, breast carcinoma and glioblastoma. This study had induced a PCP-like tumor in Rax⁺ tanocytes, but whether the pituitary cells harboring Braf mutation display the same tumorigenic potential and develop into adamantinomatous craniopharyngioma (ACP) or PCP.

Reply:

The reviewer's insight is greatly appreciated. It is well-known that the anterior pituitary derives from an invagination of the oral ectoderm known as Rathke's pouch. Lineage tracing has revealed that Hesx1-expressing cells within Rathke's pouch give rise to all the hormone-producing cells within the anterior pituitary (8-12). It has recently been shown that ectopic activation of β -catenin in Hesx1-expressing cells using Hesx1^{Cre/+}::Ctnnb1^{+lox (ex3)} mice causes hypopituitarism and ACP-like neoplasm in pituitary (6). In contrast, constitutive activation of Braf in Hesx1⁺ pituitary cells using Hesx1^{Cre/+}::Braf^{V600E} mice did not induce a substantial tumor but only caused a mild increase in cell proliferation (13). Together, these two studies have suggested that pituitary cells harboring Braf^{V600E} mutation do not develop into ACP or PCP.

Question 3:

CPs was assumed to be derived from squamous epithelial cells by clinical researchers, instead of neuroepithelial cells, despite the evidence has long been lacking. The authors would better explain and discuss the other potential origins of CPs.

Reply:

Our study together with previous studies suggested that there are 4 potential cells-of-origin for CPs: 1) Rax⁺ hypothalamic tanycytes; 2) Sox2⁺ pituitary progenitors; 3) Hesx1⁺ pituitary cells within Rathke's pouch; 4) squamous epithelial cells.

1) Our study shows that hypothalamic tanycytes, located in the human infundibulum and tuber cinereum regions, contribute to the formation of PCP-like tumor upon Braf mutation. Given the anatomic position, pathological features and gene mutation characteristic of this neoplasm, we speculate that hypothalamic tanycytes in ME serve as the cellular origin of PCPs.

2) A previous study has shown that both embryonic and adult Sox2⁺ progenitor cells in the pituitary drive the pituitary oncogenesis upon β -catenin mutation (5). Interestingly, targeted expression of oncogenic β -catenin in Sox2⁺ cells caused the formation of typical β -catenin-accumulating cell clusters, previously found in ACP, and gave rise to pituitary tumor. This study suggests that Sox2⁺ progenitor/stem cells may serve as the target cells for tumor-initiating mutations in mouse and human ACPs (5).

3) Another recent study showed that the targeted expression of oncogenic β -catenin in Hesx1⁺ embryonic pituitary precursors or Sox2⁺ pituitary stem cells in young mice induced senescence-associated secretory phenotype (SASP) and promoted the emergence of cancer stem cells and the subsequent growth of cancer cells in the pituitary niche (7). Similarly, it has been corroborated that β -catenin activation of Hesx1⁺ cells in Rathke's pouch causes hypopituitarism and ACP-like neoplasm in pituitary (6). This study implies that Hesx1-labeled progenitor cells within Rathke's pouch could also contribute to the formation of APC in a paracrine manner.

4) Previous pathological studies showed that a subpopulation of ACP and PCP tumor cells displayed the morphological characteristics of squamous epithelium in the human samples dissected from CP patients, which directed the clinicians to hypothesize that squamous epithelial cells might drive the tumor initiation in CP (14-17). However, no direct evidence has been provided as yet, neither has squamous epithelial cells been

clearly identified in the infundibular stalk or pituitary. Targeted oncogenic mutation in the potential squamous epithelial cell population within pituitary stalk is required to prove the hypothesis in the future.

Question 4:

Although the authors showed that PCP-like tumors expressed tanycyte markers such as *Ptprz1*, *Fabp7* and *Crym*, does the transcriptome of PCP-like tumors display any similarity with squamous epithelial cells (e.g. esophageal or oropharyngeal squamous epithelial cells)?

Reply:

We are grateful to the reviewer for this great suggestion. To analyze the transcriptomic similarity between PCP-like tumors and esophageal squamous cell carcinoma (ESCC), we took advantage of bulk RNAseq data collected from normal mouse esophageal squamous epithelial cells (ESECs, NOR group), inflammatory ESECs (INF group), ESCC in situ (CIS group) and invasive ESCC (ICA group) in a recently published study (18), and compared the transcriptomic profile of PCP-like tumor with ESECs or ESCCs. The principal component analysis (PCA) and correlation analysis after normalization showed that PCP-like tumors (Mutant group) resembled normal tanycytes (Control group) in their transcriptome much more than normal ESECs and the cancerous lesions (Figure R1; Supplementary Figure 10), suggesting that PCP-like tumors may not originate from squamous epithelial cells.

Figure R1. Comparative transcriptomic analysis of tanycyte- and esophageal squamous cell-derived tumors. Principal component analysis (PCA) and Spearman's correlation analysis of bulk RNAseq data collected from normal tanycytes (Control group), tumors arising from tanycytes (Mutant group), normal esophageal squamous epithelia (NOR groups), inflammatory esophageal squamous epithelial cells (INF group), in situ esophageal squamous cell carcinoma (CIS group) and invasive esophageal squamous cell carcinoma (ICA group). The RNAseq data of NOR, INF, CIS and ICA groups were downloaded from a recently published study 56. The bulk mRNA from normal cells, precancerous cells and tumor cells were collected, reverse transcribed and amplified with a similar procedure for next-generation sequencing. NOR, normal; INF, inflammation; CIS, carcinoma in situ; ICA, invasive carcinoma; M, month of mouse age.

Reference

1. M. A. Bonaguidi *et al.*, In vivo clonal analysis reveals self-renewing and multipotent adult neural stem cell characteristics. *Cell* **145**, 1142-1155 (2011).
2. E. Llorens-Bobadilla *et al.*, Single-Cell Transcriptomics Reveals a Population of Dormant Neural Stem Cells that Become Activated upon Brain Injury. *Cell Stem Cell* **17**, 329-340 (2015).
3. M. Carlén *et al.*, Forebrain ependymal cells are Notch-dependent and generate neuroblasts and astrocytes after stroke. *Nat Neurosci* **12**, 259-267 (2009).
4. B. Tang *et al.*, A novel endoscopic classification for craniopharyngioma based on its origin. *Sci. Rep.* **8**, 10215 (2018).
5. C. L. Andoniadou *et al.*, Sox2(+) stem/progenitor cells in the adult mouse pituitary support organ homeostasis and have tumor-inducing potential. *Cell Stem Cell* **13**, 433-445 (2013).
6. C. Gaston-Massuet *et al.*, Increased Wntless (Wnt) signaling in pituitary progenitor/stem cells gives rise to pituitary tumors in mice and humans. *Proc. Natl. Acad. Sci. U. S. A.* **108**, 11482-11487 (2011).
7. J. M. Gonzalez-Meljem *et al.*, Stem cell senescence drives age-attenuated induction of pituitary tumours in mouse models of paediatric

- craniopharyngioma. *Nat Commun* **8**, 1819 (2017).
8. J. Chen *et al.*, Pituitary progenitor cells tracked down by side population dissection. *Stem Cells* **27**, 1182-1195 (2009).
 9. T. Fauquier, K. Rizzoti, M. Dattani, R. Lovell-Badge, I. C. Robinson, SOX2-expressing progenitor cells generate all of the major cell types in the adult mouse pituitary gland. *Proc. Natl. Acad. Sci. U. S. A.* **105**, 2907-2912 (2008).
 10. M. Garcia-Lavandeira *et al.*, A GRFa2/Prop1/stem (GPS) cell niche in the pituitary. *PLoS One* **4**, e4815 (2009).
 11. A. S. Gleiberman *et al.*, Genetic approaches identify adult pituitary stem cells. *Proc. Natl. Acad. Sci. U. S. A.* **105**, 6332-6337 (2008).
 12. D. A. Lepore *et al.*, Identification and enrichment of colony-forming cells from the adult murine pituitary. *Exp. Cell Res.* **308**, 166-176 (2005).
 13. S. Haston *et al.*, MAPK pathway control of stem cell proliferation and differentiation in the embryonic pituitary provides insights into the pathogenesis of papillary craniopharyngioma. *Development* **144**, 2141-2152 (2017).
 14. T. Chatterjee, S. Desai, R. Lakhtakia, S. S. Gill, S. Satyanarayana, Suprasellar Papillary Squamous Epithelioma (Papillary Craniopharyngioma). *Med J Armed Forces India* **56**, 158-160 (2000).
 15. M. Chougule, in *Neuropathology of Brain Tumors with Radiologic Correlates*. (Springer Singapore, Singapore, 2020), pp. 315-322.
 16. V. C. Prabhu, H. G. Brown, The pathogenesis of craniopharyngiomas. *Childs Nerv. Syst.* **21**, 622-627 (2005).
 17. J. Zhu, C. You, Craniopharyngioma: Survivin expression and ultrastructure. *Oncol. Lett.* **9**, 75-80 (2015).
 18. J. Yao *et al.*, Single-cell transcriptomic analysis in a mouse model deciphers cell transition states in the multistep development of esophageal cancer. *Nat Commun* **11**, 3715 (2020).

Reviewers' comments:

Reviewer #1 (Remarks to the Author):

The authors addressed most of the comments to my satisfaction. The only remaining concern is Figure 3E middle (EdU/PDGFRa/tdTomato/DAPI) and bottom (EdU/S100b/tdTomato/DAPI) panels: these two IHC images remain of poor quality and non-convincing and the authors themselves admit that the heat-based antigen retrieval protocol deteriorates the tdTomato signal. Thus the two IHC images should be removed from the manuscript as they are not useful for drawing any clear conclusions.

Reviewer #2 (Remarks to the Author):

The authors have addressed the reviewers' comments. A small advice: It would have been good to include the response to the questions of reviewer 4 in the revised manuscript.

Reviewer #4 (Remarks to the Author):

The manuscript has been revised to address all of the reviewers' concerns. I think this work is highly innovative and will improve our understanding of papillary craniopharyngioma to a great extent. I support the publication of this manuscript in Nature Communication without any more concerns.

Reviewer #5 (Remarks to the Author):

This paper addresses tanycytes' regenerative and tumorigenic capacities. The authors showed that normally quiescent Rax+ tanycytes in the median eminence enter the cell cycle upon neural injury followed by self renewal and regeneration. They further demonstrated that Igf1r signaling in tanycytes is required for tissue repair under injury conditions. More interestingly, they discovered that Braf oncogenic activation is sufficient to transform Rax+ tanycytes into actively dividing tumor cells that eventually develop into a papillary craniopharyngioma-like tumor.

While this paper is highly interesting and well-executed, two major problems arise from one main related issue, that is the number of Rax-positive cells from 10 mouse was only around 900 cells. This perhaps led to the lack of resolution for the previously known four subtypes of Rax-positive tanycytes (alpha 1, alpha 2, beta 1 and beta 2 tanycytes). This paper will be much stronger if the authors provide the following two lines of further clarification/data.

1) While the cell number is low, I wonder if they still tried to further sub-cluster the Rax+ cell cluster to see if it can get separated to the four subtypes. At the very least, the reason for the low number of single cells analyzed should be discussed.

2) More importantly, it will be very interesting to determine which tanycyte subtype is responsible for self renewal/regeneration and tumorigenesis, respectively. This can be relatively easily done using the known markers of each subtype.

Reviewer #1 (Remarks to the Author):

The authors addressed most of the comments to my satisfaction. The only remaining concern is Figure 3E middle (EdU/PDGFRa/tdTomato/DAPI) and bottom (EdU/S100b/tdTomato/DAPI) panels: these two IHC images remain of poor quality and non-convincing and the authors themselves admit that the heat-based antigen retrieval protocol deteriorates the tdTomato signal. Thus the two IHC images should be removed from the manuscript as they are not useful for drawing any clear conclusions.

Reply:

We sincerely appreciate the reviewer's comment and the editor's suggestion. We would like to take the editor's suggestion to keep the images in Figure 3 and further explained the limitations of our approach in the text (Page 7, Paragraph 2). We have also toned down our conclusion as follows:

The results of fate mapping using Rax-CreERT2::Ai14 mice further showed that neural injury induced a predominant self-renewal of tanycytes and drove a tiny minority of tanycytes to differentiate into OPCs or astrocytes (Fig. 3e and Supplementary Fig. 5b-d), implicating the multipotential differentiation of activated tanycytes into glial cells. Notably, given that our heat-based antigen retrieval approach deteriorated the tdTomato signal to a certain extent, whether ME tanycytes robustly differentiate into OPCs or astrocytes upon injury requires further confirmation.

Reviewer #2 (Remarks to the Author):

The authors have addressed the reviewers' comments. A small advice: It would have been good to include the response to the questions of reviewer 4 in the revised manuscript.

Reply:

We thank the reviewer's great advice. Actually, we have succinctly replied to the questions from Reviewer #4 in the previous version of manuscript. Despite that

Reviewer #4 asked 4 questions, his/her main concerns focused on the potential origins of craniopharyngioma. We have included the following response and 1 Supplementary Figure in the revised manuscript (Page 13, Paragraph 3; Page 14, Paragraph 1):

Craniopharyngioma is a heterogeneous brain tumor of uncertain origin and its tumor biology is poorly understood. Recent studies show that pituitary stem/progenitor cells carrying Cnntb1 but not Braf mutation contribute to the development of pituitary tumors and/or adamantinomatous craniopharyngioma via paracrine mechanism. Interestingly, our results reveal that tanycyte-derived tumors in ME mimic papillary craniopharyngioma (frequently carrying BrafV600E mutation) with respect to their anatomic location, genetic mutation and pathological features, suggesting that hypothalamic Rax+ tanycytes could serve as a cell-of-origin for papillary craniopharyngioma. In contrast to the salient OPC features in glioma, the tumor cells in our disease model do not abundantly express differentiated cell markers such as Olig2, S100 β and CC1. Notably, craniopharyngioma was previously assumed to derive from squamous epithelial cells by clinical researchers, but we did not find a robust transcriptomic similarity between tanycytes and esophageal squamous epithelia, or between tanycytes-derived tumor and esophageal squamous cell carcinoma (Supplementary Fig. 10). The data from our transcriptomic profiling further demonstrate that tanycyte-derived tumors display an enrichment of genes involved in brain but not pituitary development.

Reviewer #4 (Remarks to the Author):

The manuscript has been revised to address all of the reviewers' concerns. I think this work is highly innovative and will improve our understanding of papillary craniopharyngioma to a great extent. I support the publication of this manuscript in Nature Communication without any more concerns.

Reply:

We appreciate the reviewer's favorable comments on our work.

Reviewer #5 (Remarks to the Author):

While this paper is highly interesting and well-executed, two major problems arise from one main related issue, that is the number of Rax-positive cells from 10 mouse was only around 900 cells. This perhaps led to the lack of resolution for the previously known four subtypes of Rax-positive tanycytes (alpha 1, alpha 2, beta 1 and beta 2 tanycytes). This paper will be much stronger if the authors provide the following two lines of further clarification/data.

Question 1:

1) While the cell number is low, I wonder if they still tried to further sub-cluster the Rax+ cell cluster to see if it can get separated to the four subtypes. At the very least, the reason for the low number of single cells analyzed should be discussed.

Reply:

We deeply understand the reviewer's concern and apologize for any confusion this may cause. Here we would like to provide detailed interpretations to clarify the reviewer's concern. Firstly, given a previous study showing that ME tanycytes might serve as a third population of postnatal neural stem cells (1), the goal of our study is to reveal the regenerative and tumorigenic potential of tanycytes in median eminence (ME). Thus, we only microdissected ME region (containing $\beta 2$ tanycytes) for single-cell RNAseq (Fig. R1) and were devoted to uncover the general marker and properties of ME tanycytes using our single-cell dataset. As such, our single-cell dataset would not cover all of the previously known four subtypes of tanycytes but only $\beta 2$ subtype to a large extent.

Fig. R1. Spatial distribution of 4 subtypes of tanycytes along the 3rd ventricle. **a**, Schematic diagram of coronal mouse brain section. **b-c**, The spatial distribution of $\alpha 1$, $\alpha 1$, $\beta 1$ and $\beta 2$ tanycytes along the 3rd ventricle. Median eminence (ME) contains $\beta 2$ tanycytes only.

Secondly, the single-cell database in the manuscript is indeed relatively small, but a larger database covering different age, gender, mouse lines and species is cumulatively expanding in our lab. Given that the quality of this single-cell dataset is very good (Fig. R2a) and single-cell taxonomy of tanycytes is not our key point in the manuscript, we still would like to keep the 990-cell dataset for further analyses.

Thirdly, we followed the reviewer's suggestion to make a further sub-clustering of the Rax^+ tanycytes, examined the expression of subtype markers identified by Campbell and his colleagues (2), and found that the collected tanycytes were composed of a majority of $\beta 2$ subtype (marked by *Scn7a* and *Col25a1*) and a minority of boundary cells between $\beta 1$ and $\beta 2$ subtype (marked by coexpression of *Scn7a* and *Gria2*), but there were almost no $\alpha 1$ and $\alpha 2$ subpopulations (Fig. R2b-2i).

Fig. R2. Subclustering of *Rax*⁺ tanycytes collected from median eminence (ME). **a**, Quality evaluation of single-cell dataset. **b**, tSNE subclustering of ME tanycytes. **c**, Heatmap showing the differentially expressed genes of tanycyte subtypes in ME. **d**, Heatmap showing the expression of previously identified ependymal cells (Epy), $\alpha 1$, $\alpha 2$, $\beta 1$ and $\beta 2$ subtype markers in ME tanycytes (2). **e-g**, Feature plots showing the marker gene expression (*Col25a1*, *Scn7a* and *Rax*) in ME tanycytes. **h**, *Gria2* expression in $\beta 1$ tanycytes shown by *in situ* hybridization of coronal brain sections (Allen Brain Atlas). **i**, Feature plot showing the expression of *Gria2* in the c3 subcluster of ME tanycytes.

Lastly, to be honest, due to the tininess of ME tissue and the lack of experience in dissociating such tiny tissues back to 2017, we only acquired a relatively small dataset for single-cell analysis. Despite the low number of collected cells, we still provided high-quality data. As aforementioned, we have been collecting a larger database covering different age, gender, mouse lines and species, which will be published and released in the near future. For example, in one of our ongoing projects, we also revealed 4 subtypes of tanycytes arising from *Tbx3*-positive hypothalamic progenitors during development (Fig. R3, Unpublished data). But even so, we would like to take the reviewer's suggestion to discuss the reasons for the low number of collected cells in the "Methods and Materials" Section (Page 16, Paragraph 1).

Fig. R3. *Tbx3*-derived tanycytes in hypothalamus. **a**, tSNE clustering of tanycytes arising from *Tbx3*⁺ neural progenitors. We applied a single dose of tamoxifen at E9 in *Tbx3*-*CreER*^{T2};Ai14 mice and sorted the tdTomato⁺ cells in hypothalamus for single-cell RNAseq. **b**,

Dot plot showing the specific markers for different tanycyte clusters. **c**, Expression of tdTomato in tanycytes lining the third ventricle (left) and the marker gene expression shown by in situ hybridization of coronal brain sections (Allan Mouse Brain Atlas).

Question 2:

2) More importantly, it will be very interesting to determine which tanycyte subtype is responsible for self renewal/regeneration and tumorigenesis, respectively. This can be relatively easily done using the known markers of each subtype.

Reply:

We thank the reviewer for this great suggestion. Indeed, it will be intriguing to determine the regenerative and tumorigenic potentials of different subtype of tanycytes. However, given that our current study focuses on the cellular properties of β tanycytes labeled by Rax-CreER^{T2} mouse line (Fig. R4; Supplementary Fig. 4), especially $\beta 2$ tanycytes in ME, the regenerative and tumorigenic potentials of α tanycytes remain elusive to us.

Fig. R4. Lineage tracing of tanycytes using Rax-CreER^{T2} knockin mouse line. a, Representative images showing the genetic labeling of tanycytes lining the third ventricle using Rax-CreER^{T2}::Ai14 mouse line at 1 day post induction (dpi) with tamoxifen. Scale bar, 1 mm. **b**, Predominant labeling of $\beta 1$ and $\beta 2$ tanycytes by Rax-CreER^{T2} tool mouse line.

Even when we tried to resolve Rax^+ tanycytes into $\beta 1$ and $\beta 2$ subtypes, we found that it remains difficult to distinguish these two subtypes using unique markers (Fig. R5a-R5c). Previous literatures convincingly suggested that $Fgf10$ and Rax predominantly label β tanycytes, while $Rarres2$ marks α tanycytes and ependymal cells along the 3rd ventricle (2-5). Nevertheless, there was no well-recognized markers hitherto to specifically distinguish $\beta 1$ and $\beta 2$ subtypes. Despite a recent molecular census of arcuate hypothalamus and ME suggesting that $Scn7a$ may serve as a specific marker for $\beta 2$ tanycytes (Fig. R5d), our single-molecule fluorescent *in situ* hybridization (smFISH) results indicated that $Scn7a$ was expressed in both $\beta 1$ and $\beta 2$ subtypes (Fig. R5e and R5f; Fig. 1e; Supplementary Fig. 4b).

a and b adapted from Goodman & Hajhosseini, 2015

c and d adapted from Campbell, Nat Neuro, 2017

Fig. R5. There is no well-recognized markers hitherto to specifically distinguish $\beta 1$ and $\beta 2$ tanycytes. **a**, Organization and distribution of hypothalamic tanycytes . **b**, Common and domain-restricted gene expression within the cell lining of the adult 3rd ventricle . **c**, Violin plots showing the previously identified tanycyte subtype markers (2). **d**, Violin plots showing the newly identified tanycyte subtype markers (2). **e**, Representative images of traced tdTomato⁺ cells stained for tanycyte marker Scn7a in Rax-CreER^{T2}::Ai14 mice at 1 day post tamoxifen induction. **f**, Three-dimensional (3D) simulation of Scn7a-positive signals in ME.

Despite all of these limitations, we still appreciate the reviewer's question. Given the lack of well-recognized markers to distinguish $\beta 1$ and $\beta 2$ subtypes, we would like to discuss their regenerative and tumorigenic potentials based on their spatial position in order to satisfy the reviewer. As compared with control, genetic injury of Rax⁺ tanycytes induced the mitotic division of both $\beta 1$ and $\beta 2$ tanycytes, while mechanical injury without cell type specificity caused the proliferation of both α and β tanycytes lining the 3rd ventricle (Fig. R6a-R6c). Interestingly, although we observed the mitotic division of both $\beta 1$ and $\beta 2$ tanycytes upon oncogenic activation of Braf at 7 days post induction (dpi, Fig. R6d), it seems that only $\beta 2$ tanycytes transformed into tumor cells and depleted themselves at 30 dpi whereas $\beta 1$ tanycytes largely maintained themselves (Fig. R6e). Collectively, we speculate that both $\beta 1$ and $\beta 2$ tanycytes are capable of self-renewal for regeneration, but $\beta 2$ tanycytes are more susceptible to oncogenic activation.

Fig. R6. Regenerative and tumorigenic potentials of tanycytes. **a**, Representative images showing the EdU⁺ mitotic cells distributing in mediobasal hypothalamus (MBH). Scale bar, 100 μ m. **b**, Self-renewal of residual β tanycytes after partial genetic ablation of Rax⁺ tanycytes. Arrowheads indicate the mitotic cells in β 1 subpopulation. **c**, Mitotic activation of α and β tanycytes lining the 3rd ventricle. Arrowheads indicate the mitotic cells in α subpopulation. **d**, Oncogenic activation of Braf incites the expansion of β tanycytes at 7 days post induction (dpi). Arrowheads indicate the mitotic cells in β 1 subpopulation. **e**, β 2 tanycytes display tumorigenic potential and deplete themselves at 30 dpi, while β 1 tanycytes maintain themselves. Scale bar, 100 μ m.

We have added a paragraph to emphasize that our study focuses on the regenerative and tumorigenic potentials of β tanycytes in our “Discussion” Section (Page 14, Paragraph 2) as follows:

Given that previous studies have subdivided tanycytes into α 1, α 2, β 1 and β 2 subtypes along the 3rd ventricle, here we have to emphasize that our study focuses on Rax⁺ tanycytes, predominantly composed of β 1 and β 2 subtypes. However, the regenerative and tumorigenic potentials of α tanycytes remain elusive to us.

Reference:

1. D. A. Lee *et al.*, Tanycytes of the hypothalamic median eminence form a diet-responsive neurogenic niche. *Nat Neurosci* **15**, 700-702 (2012).
2. J. N. Campbell *et al.*, A molecular census of arcuate hypothalamus and median eminence cell types. *Nat Neurosci* **20**, 484-496 (2017).
3. T. Goodman, M. K. Hajihosseini, Hypothalamic tanycytes-masters and servants of metabolic, neuroendocrine, and neurogenic functions. *Front Neurosci* **9**, 387 (2015).
4. A. L. Miranda-Angulo, M. S. Byerly, J. Mesa, H. Wang, S. Blackshaw, Rax regulates hypothalamic tanycyte differentiation and barrier function in mice. *J Comp Neurol* **522**, 876-899 (2014).
5. N. Haan *et al.*, Fgf10-expressing tanycytes add new neurons to the appetite/energy-balance regulating centers of the postnatal and adult hypothalamus. *J Neurosci* **33**, 6170-6180 (2013).

Reviewers' comments:

Reviewer #1 (Remarks to the Author):

Just one, very minor note:

To more appropriately reflect the technical issue at hand, I'd suggest to re-phrase the first sentence as follows:

"...neural injury induces a predominant self-renewal of tanycytes and may drive a small number of tanycytes to differentiate into OPCs or astrocytes..."

Reviewer #5 (Remarks to the Author):

The authors very nicely addressed this reviewer's critiques supported by their unpublished data. With this revision, I think this manuscript stands much stronger and I recommend an immediate publication of this highly interesting and important paper in Nature Comm.

Response to Reviewer #1 (Remarks to the Author):

Just one, very minor note:

To more appropriately reflect the technical issue at hand, I'd suggest to re-phrase the first sentence as follows:

"...neural injury induces a predominant self-renewal of tanycytes and may drive a small number of tanycytes to differentiate into OPCs or astrocytes..."

Reply:

We sincerely appreciate the reviewer's comment. We followed the reviewer's suggestion and revised the related text as follows: "The results of fate mapping using Rax-CreERT2::Ai14 mice further showed that neural injury induced a predominant self-renewal of tanycytes and drove a small number of tanycytes to differentiate into OPCs or astrocytes" (Page 7). The change was highlighted in red.